# Topological holography, quantum criticality, and boundary states

Sheng-Jie Huang[1] and Meng Cheng[2]

[1] *Max Planck Institute for the Physics of Complex Systems,*
*Nöthnitzer Str. 38, 01187 Dresden, Germany*
[2] *Department of Physics, Yale University, New Haven, CT 06520-8120, USA*
(Dated: November 7, 2023)

Topological holography is a holographic principle that describes the generalized global symmetry of a local quantum system in terms of a topological order in one higher dimension. This framework separates the topological data from the local dynamics of a theory and provides a unified description of the symmetry and duality in gapped and gapless phases of matter. In this work, we develop the topological holographic picture for $(1+1)d$ quantum phases, including both gapped phases as well as a wide range of quantum critical points, including phase transitions between symmetry protected topological (SPT) phases, symmetry enriched quantum critical points, deconfined quantum critical points and intrinsically gapless SPT phases. Topological holography puts a strong constraint on the emergent symmetry and the anomaly for these critical theories. We show how the partition functions of these critical points can be obtained from dualizing (orbifolding) more familiar critical theories. The topological responses of the defect operators are also discussed in this framework. We further develop a topological holographic picture for conformal boundary states of $(1+1)d$ rational conformal field theories. This framework provides a simple physical picture to understand conformal boundary states and also uncovers the nature of the gapped phases corresponding to the boundary states.

## CONTENTS

## I. INTRODUCTION

Classification and characterization of phases of matters has been a central theme in condensed matter physics. Traditionally, phases and phase transitions have been described by Landau symmetry breaking theory. However, recent developments in the field have led to the discovery of novel phases of matter that cannot be described by Landau theory. As a prominent example of phases beyond Landau paradigm, there has been a significant progress in the study of gapped quantum phases, including topological orders with long-rang entanglement[1] and various topological phases distinguished by symmetry, such as symmetry-enriched topological (SET)[2–4] or symmetry-protected topological (SPT) phases[5], depending on whether there are non-trivial fractionalized excitations.

Much less is known for the gapless phases and phase transitions. It has been found that various kinds of non-Landau transitions may occur in quantum systems, including the deconfined quantum critical points (DQCPs)[6], where Landau-forbidden direct transitions are possible between two phases, in which the symmetry groups on the two sides do not have the group-subgroup relation, as well as transitions between topological phases, which can not possibly be described in terms of fluctuating local order parameters. It is thus desirable to have a general theoretical framework to describe gapless phases and critical points. Although conformal field theories (CFTs) provide a powerful framework for a large class of gapless states, it is however a non-trivial task to identify the correct CFT description of a phase transition.

Symmetry plays a very important role in our understanding of phase of matters. It offers useful guidance of classifications and also leads to important physical implications, such as conservation laws, and constraints on low-energy dynamics. It has been well-known that there is an intimate relation between the global symmetry described by a symmetry group $G$ and codimension-1 topological defects satisfying group multiplication fusion rules. Building on this understanding, the notion of symmetry has been vastly generalized, so every topological defect can be viewed as a generalized global symmetry[7–9]. Even though a full description of a quantum phase includes many dynamical details, the description of topological defects can be purely algebraic and is believed to have a higher categorical description[10–18].

Recently, a topological holographic framework has emerged, which provides a holographic viewpoint of symmetry. The essential idea is to encodes the symmetry data in terms of a topological order that lives in one higher dimension[18–41], which generalizes older development in the context of $(1+1)d$ rational CFTs[42–45]. This topological order is commonly called a *symmetry topological order* or *categorical symmetry* in the condensed matter literature[23, 25, 29, 34–36], or a *symmetry TFT* in the high-energy literature[28, 32, 38, 40, 41, 46, 47]. This approach can describe both gapped and gapless phases in a unified framework, which essentially decouples the dynamics of a quantum system from the symmetry we would like to study, and allows one to make non-perturbative statements about phases, phases transitions and dualities.

In this work, we develop topological holographic picture for $(1+1)d$ quantum systems and boundary states in CFTs. In section II, we review the main idea of topological holography. The central picture is given by the "sandwich" construction, where we can view the $(1+1)d$ quantum system of interest as a slab of $(2+1)d$ topological order with appropriately chosen boundary conditions. The left boundary is always chosen to be a gapped topological boundary, and the confined anyon lines on the boundary implement the global symmetry of the original $(1+1)d$ system. We choose the right boundary condition such that when we compactify the whole slab, it produces the $(1+1)d$ quantum phase we would like to study. Choosing a gapped boundary on

the right corresponds to realizing a $(1+1)d$ gapped phase as a sandwich. This picture can describe conformal field theories by choosing some non-topological boundary condition on the right (Sec. II D) and also non-trivial gapless phases such as intrinsically gapless SPT phases (Sec. IV D). In this paper, we work with the anyon basis in contrast to the basis labeled by the flat connections. The algebraic theory and intuitions developed in topological orders can be readily applied in our approach.

In this representation, local operators are organized into different classes labeled by the anyon lines connecting the two boundaries, and the symmetry acts on the local operators through mutual braiding with the confined anyons on the left boundary. Non-local defect operators are also represented as anyon lines connecting the two boundaries with a "tail" confining on the left boundary. This picture explicitly reveals the *dual symmetry* which acts trivially on all the local operators and only acts non-trivially on the non-local defect operators.

We summarize the key contributions of this work as follows:

- We discuss $(1+1)d$ gapped phases with modular category symmetry and with the usual group symmetry in the framework of topological holography in section III. We demonstrate through many examples that the correspondence between the topological gapped boundary conditions in the sandwich picture and the $(1+1)d$ gapped phases. For spontaneous symmetry breaking phases, we explicitly construct the order parameters in terms of the bulk anyon lines that connecting the two gapped boundaries. For SPT phases, we also discuss the symmetry defect operators and the relation to the protected edge degeneracy.

- We discuss $(1+1)d$ gapless phases and critical points in the framework of topological holography in section IV. Dualities[48] in $(1+1)d$ systems can also be systematically studied in the framework of topological holography. In general, there are two classes of dualities. For the first class, they correspond to inserting twist defects that permute anyons in the sandwich picture. When the defect satisfying a $\mathbb{Z}_2$ fusion rule, it corresponds to a self-duality. The other class corresponds to simply choosing a different topological gapped boundary conditions on the left, which is not necessarily related to the old one by anyon permutation. This topological holographic viewpoint of dualities is very powerful. It allows one to relates the partition functions between different quantum critical points and sometimes one can explicitly obtain the partition function of an exotic critical point by dualizing it to a more familiar critical theory. We will see many examples in section IV, including some exotic quantum critical points such as phase transition between SPT phases, symmetry enriched quantum critical points, and DQCPs.

- Many of these exotic quantum critical points and the intrinsically gapless SPT phases can be characterized by the non-trivial responses in the symmetry defect operators. These responses serve as the topological invariants and can be obtained by calculating the twisted partition functions. In section IV, we also discuss these topological invariants and the twisted partition functions in the framework of topological holography. These topological invariants are conveniently encoded through non-trivial braidings of anyons in the symmetry topological orders in the bulk.

- $(1+1)d$ gappled phases can be obtained by turning on some relevant perturbations in a $(1+1)d$ CFT. Studying the fate and the renormalization group (RG) flows of perturbed $(1+1)d$ CFTs is an important but difficult question. A useful tool to study such flow is through studying boundary conditions, which is obtained by starting with a CFT on a line and perturbing it by a relevant operator on a half-line. We consider the case where the relevant perturbation drives the CFT into a gapped phase. The interface between the CFT and the resulting gapped phase is described by a conformal boundary condition, which is usually called an RG boundary. If the perturbed theory is trivially gapped, the RG boundary is elementary (irreducible), described by the Cardy states (or the non-Cardy states for non-diagonal CFTs). If there is a vacuum degeneracy, the RG boundary is in general a superposition of elementary conformal boundaries. Therefore, for each gapped phase, we can identify a set of elementary conformal boundary conditions appearing in the RG boundary and thus establish the correspondence between the gapped phases and the boundary states.

  We show in section V that the topological holography gives a simple physical picture to describe conformal boundary states of $(1+1)d$ rational CFTs. Specifically, the boundary states correspond to the quantum quench process where the edge chiral CFTs of the bulk topological order are coupled to form a gapped phase. This picture can describe the traditional Cardy states as well as many non-Cardy boundary states in non-diagonal CFTs. We use the topological holography picture to derive boundary

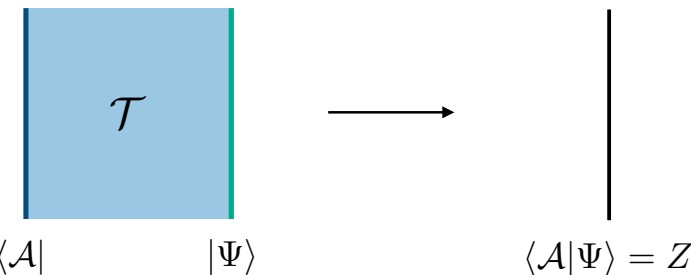

FIG. 1. The sandwich picture in the topological holography. A $(1+1)d$ theory can be viewed as a sandwich built from a topological order $\mathcal{T}$ in the bulk with a topological gapped boundary conditions on the left and a potentially non-topological boundary condition on the right.

CFT partition functions in section V C. In section V E, we demonstrate through many examples that the correspondence between the gapped phases and the boundary states can be achieved by using the physical picture derived from the topological holography together with the help of Cardy's variational argument[49].

## II. TOPOLOGICAL HOLOGRAPHY AND THE SANDWICH CONSTRUCTION

Here we give a brief overview of the topological holography, focusing on $(1+1)d$. The essential idea of the topological holography is to encode the symmetry data of a $(1+1)d$ quantum phase of matter into a *sandwich*, which is a $(2+1)d$ topological order $\mathcal{T}$ defined on a strip with appropriate boundary conditions. One of the main benefits of this approach is to separate the topological data of the symmetry from the potentially non-topological contents in the $(1+1)d$ theory of interest. Fig. 1 illustrates the main idea. The left boundary is chosen to be a topological gapped boundary of the $(2+1)d$ topological order, which encodes all the topological data of the symmetry. Fixing a left boundary, different choices of the right boundary result in different $(1+1)d$ quantum phases. We note that the right boundary could be non-topological, such as a CFT. We will sometimes call the left topological boundary as the *reference boundary*, since different $(1+1)d$ phases can be analysed on the same footing only when we fix a choice of the left boundary. As we will discuss below, changing the left boundary corresponds to a duality transformation of the $(1+1)d$ quantum system.

### A. Boundary states and partition functions

To be more concrete, consider the Euclidean $(2+1)d$ theory on an open 3D manifold $M \times I$, where $I = [0, 1]$ is a 1D interval, and $M$ is a closed surface. We take $M \times \{0\}$ as the left boundary, and $M \times \{1\}$ as the right boundary. The partition function on this $(2+1)d$ manifold can be viewed as the inner product of two states:

$$Z = \langle \mathcal{A} | \Psi \rangle, \tag{1}$$

where $|\mathcal{A}\rangle$ is the left (topological) boundary state, and $|\Psi\rangle$ is the right boundary state. When the surface $M$ is not a sphere (i.e. nonzero genus), the Hilbert space of the bulk topological theory on $M$ has dimension greater than 1, and we choose a (orthonormal) basis for the Hilbert space labeled as $|\alpha\rangle$. There is a canonical choice for the basis states $|\alpha\rangle$. When $M$ is a torus, the label $\alpha$ corresponds to an anyon type in the bulk, and the state $|\alpha\rangle$ can be obtained as the basis of the Hilbert space on the solid torus $D^2 \times S^1$, where $D^2$ is a disk, with an insertion of anyon $\alpha$ wrapping around $S^1$.

The right boundary state $|\Psi\rangle$, which may or may not be topological, can always be expanded in the anyon basis $|\alpha\rangle$:

$$|\Psi\rangle = \sum_{\alpha \in \mathcal{T}} Z_\alpha |\alpha\rangle. \tag{2}$$

When $M$ is a torus $T^2$, the coefficient $Z_\alpha$ in this expansion is equal to the partition function defined on the solid torus $D^2 \times S^1$ with $\Psi$ boundary condition and an insertion of anyon $\alpha$ around $S^1$.

In order for the sandwich construction to produce a $(1+1)d$ theory, we must choose a topological gapped boundary condition on the left boundary. Gapped boundaries of a $(2+1)d$ topological order are classified by the Lagrangian algebra $\mathcal{A} = \bigoplus_{\alpha \in \mathcal{T}} w_\alpha \alpha$, where $w_\alpha$ are some non-negative integers. Each topological gapped boundary corresponds to a state

$$|\mathcal{A}\rangle = \sum_{\alpha \in \mathcal{T}} w_\alpha |\alpha\rangle. \tag{3}$$

The Lagrangian algebra needs to satisfy certain consistency conditions, which will be briefly reviewed in Appendix. Physically, the Lagrangian algebra describes condensation of anyons on the boundary: if $w_\alpha > 0$, then an anyon $\alpha$ can condense on the boundary.

The sandwich construction is essentially a dimensional reduction of a $(2+1)d$ topological order with appropriate choices of boundary conditions such that, after the dimensional reduction, we recover the $(1+1)d$ phase of interest. If $M$ is a torus $T^2$, we can also shrink the topological boundary of the sandwich and obtain a $(2+1)d$ topological order with an insertion of Lagrangian algebra $\mathcal{A}$ as shown in Fig. 2. Now the partition function of the $(1+1)d$ theory can be computed by taking the inner product:

$$\begin{aligned} Z &= \langle \mathcal{A} | \Psi \rangle \\ &= \sum_{\alpha, \beta \in \mathcal{T}} w_\beta Z_\alpha \langle \beta | \alpha \rangle \\ &= \sum_{\alpha \in \mathcal{T}} w_\alpha Z_\alpha. \end{aligned} \tag{4}$$

Therefore, the partition function of the $(1+1)d$ theory can be written as a linear combination of the partition function of the topological order $\mathcal{T}$ on the solid torus with an insertion of of Lagrangian algebra $\mathcal{A}$ around $S^1$. The construction can be easily generalized to higher genus surfaces to compute the partition functions.

We now discuss two examples that will be important throughout the work.

Suppose $G$ is a finite group, and consider the bulk topological order to be a twisted quantum double $\mathcal{Z}(\text{Vec}_G^\omega)$, where $\omega \in \mathcal{H}^3(G, \text{U}(1))$. Physically the bulk can also be viewed as a (twisted) $G$ gauge theory, so there is a subcategory of gauge charges isomorphic to $\text{Rep}(G)$ (i.e. the category of irreducible linear representations of $G$). The canonical boundary is the one where all gauge charges condense, and the gauge symmetry becomes a physical symmetry $G$ on the boundary. The Lagrangian algebra $\mathcal{A}$ is given by

$$\mathcal{A} = \sum_{\alpha \in \text{Rep}(G)} d_\alpha \alpha, \tag{5}$$

where $\alpha$ runs over all irreps of $G$. Note that when $\omega$ is nontrivial, the $G$ symmetry in this $(1+1)d$ system is anomalous.

For another example, suppose $\mathcal{B}$ is a modular tensor category. It is well-known that $\mathcal{Z}(\mathcal{B}) = \mathcal{B} \boxtimes \bar{\mathcal{B}}$. The canonical boundary condition is given by the following Lagrangian algebra:

$$\mathcal{A} = \sum_{a \in \mathcal{B}} (a, a). \tag{6}$$

Here $(a, a')$ labels anyons in the bulk $\mathcal{Z}(\mathcal{B})$.

### B. Symmetries and operators

Below we will discuss another aspect of the holography, namely the operator content, including both local operators and topological defect lines.

To this end, it is useful to consider the spacetime picture as shown in Fig. 3. In the sandwich construction, the topological defect lines confined on the left boundary implement the symmetry of our system. In general, the topological defect lines are described by a fusion category $\mathcal{C}$[50]. The anyons in the bulk topological order are then described by a modular tensor category called the Drinfeld center $\mathcal{Z}(\mathcal{C})$.

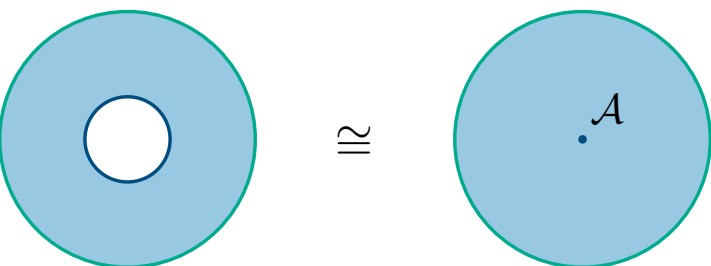

FIG. 2. When $M$ is a torus $T^2$, we can shrink the inner topological boundary of the sandwich $T^2 \times I$, and it becomes a Lagrangian algebra given by $\mathcal{A}$. The figure shows a spatial slice $S^1 \times I$ of this process.

Local operators on the right boundary can be organized according to their transformations under the fusion category symmetry. As a result, a local operator can be uniquely attached a bulk anyon label $\alpha$ with $w_\alpha > 0$. Intuitively, the local operator $\mathcal{O}_\alpha$ corresponds to a bulk anyon line $\alpha$ that condenses on the left topological boundary, and stretches across the sandwich, as shown in Fig. 4(a). When $\alpha$ is an non-abelian anyon, there may be multiple condensation channels (splitting idempotents[51]) on the left boundary. Each condensation channel $i$ corresponds to an local operator $\mathcal{O}_\alpha^i$.

The symmetry $\mathcal{C}$ acts on the local operator as the *half braiding* between the bulk anyon line $\alpha \in \mathcal{Z}(\mathcal{C})$ and the boundary topological line $m \in \mathcal{C}$, which is defined as follows:

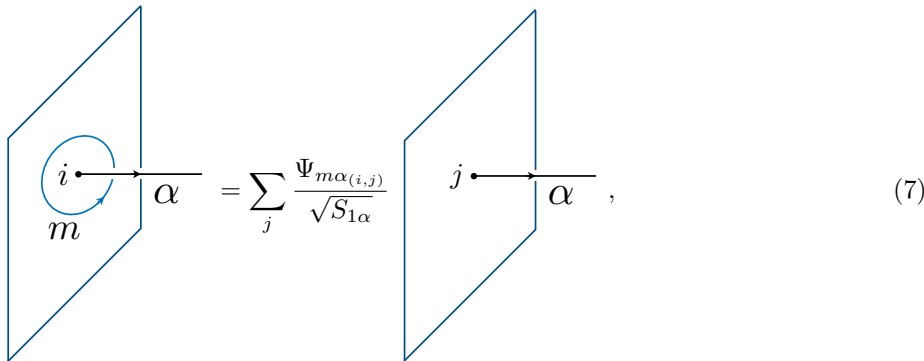

$$\tag{7}$$

where $i, j$ label the condensation channels, $S_{1\alpha}$ is the S-matrix element of the bulk anyons, and $\Psi_{m\alpha_{(i,j)}}$ is the *half-braiding matrix*. We denote the symmetry action as

$$U_m \cdot \mathcal{O}_\alpha^i = \sum_j \frac{\Psi_{m\alpha_{(i,j)}}}{\sqrt{S_{1\alpha}}} \mathcal{O}_\alpha^j. \tag{8}$$

For more detailed discussion of the half-braiding (also called the half-linking), please see Ref. [52, 53]. Here we follow the convention in Ref. [53]. In this convention, we have

$$\Psi_{11} = \sqrt{S_{11}}, \ \ \Psi_{a1} = \sqrt{S_{11}}d_a, \ \ \Psi_{1\alpha_{i,j}} = \delta_{ij}\sqrt{S_{1\mu}}. \tag{9}$$

When the symmetry is given by a symmetry group $G$, the topological defect lines on the reference boundary are labeled by group elements, and the fusion satisfies the group multiplications (described by $\text{Vec}_G$). The local operators correspond to the bulk anyon lines that can condense on the reference boundary. When we choose the reference boundary to be the charge condensed boundary, the local operators corresponds to the pure charges in the quantum double $D(G)$, labeled by $([1], \pi)$. Here $[1]$ denotes the conjugacy class of the identity element and $\pi$ is an irrep of the centralizer of $[1]$, which equals to the group $G$. The Lagrangian algebra is then

$$\mathcal{A} = \sum_\pi d_\pi([1], \pi), \tag{10}$$

where $d_\pi$ is the dimension of the irrep $\pi$. This agrees with the usual notion of global symmetry, where operators can be organized by linear irreducible representations of the symmetry group.

Therefore, to obtain the symmetry action, we need the half-braiding matrix between the group elements and irreps of $G$, which is provided in Ref. [52]:

$$\Psi_{gR_{ij}} = \sqrt{\frac{d_R}{|G|}} \pi_{ij}^R(g^{-1}). \tag{11}$$

Let's consider an example given by the spontaneous symmetry breaking phase of $S_3 = \{r, s | r^3 = s^2 = srsr = 1\}$. The bulk of the sandwich is a $S_3$ gauge theory. The $S$-matrix of the $S_3$ gauge theory is

$$S = \frac{1}{6} \begin{pmatrix} 1 & 1 & 2 & 3 & 3 & 2 & 2 & 2 \\ 1 & 1 & 2 & -3 & -3 & 2 & 2 & 2 \\ 2 & 2 & 4 & 0 & 0 & -2 & -2 & -2 \\ 3 & -3 & 0 & 3 & -3 & 0 & 0 & 0 \\ 3 & -3 & 0 & -3 & 3 & 0 & 0 & 0 \\ 2 & 2 & -2 & 0 & 0 & 4 & -2 & -2 \\ 2 & 2 & -2 & 0 & 0 & -2 & -2 & 4 \\ 2 & 2 & -2 & 0 & 0 & -2 & 4 & -2 \end{pmatrix}. \tag{12}$$

Here the anyon types are labeled by $A$ to $H$ alphabetically. $A$ is the identity, $B$ is the sign irrep of $S_3$, $C$ is the two-dimensional irrep. $D$ and $E$ both correspond to the conjugacy class $\{s, sr, sr^2\}$, whose centralizer is $\mathbb{Z}_2$. $D$ ($E$) carries a trivial (non-trivial) rep. under $\mathbb{Z}_2$. $F, G, H$ correspond to the conjugacy class $\{r, r^2\}$, whose centralizer is $\mathbb{Z}_3$.

To describe the SSB phase that breaks the $S_3$ symmetry completely, we choose the charge condensed boundary $\mathcal{A} = A \oplus B \oplus 2C$ on both boundaries of the sandwich. The half-braiding matrix can be obtained by Eq. (11)[52]:

$$\Psi^{S_3} = \frac{1}{\sqrt{6}} \begin{pmatrix} 1 & 1 & \sqrt{2} & 0 & 0 & \sqrt{2} \\ 1 & -1 & \sqrt{2} & 0 & 0 & -\sqrt{2} \\ 1 & 1 & -\frac{1}{\sqrt{2}} & -\sqrt{\frac{3}{2}} & \sqrt{\frac{3}{2}} & -\frac{1}{\sqrt{2}} \\ 1 & -1 & -\frac{1}{\sqrt{2}} & -\sqrt{\frac{3}{2}} & -\sqrt{\frac{3}{2}} & \frac{1}{\sqrt{2}} \\ 1 & 1 & -\frac{1}{\sqrt{2}} & \sqrt{\frac{3}{2}} & -\sqrt{\frac{3}{2}} & -\frac{1}{\sqrt{2}} \\ 1 & -1 & -\frac{1}{\sqrt{2}} & \sqrt{\frac{3}{2}} & \sqrt{\frac{3}{2}} & \frac{1}{\sqrt{2}} \end{pmatrix}, \tag{13}$$

where we use the following basis for the group element: $\{1, s, r, sr, r^2, sr^2\}$, and the basis for the irreps: $\{A, B, C_{(1,1)}, C_{(1,2)}, C_{(2,1)}, C_{(2,2)}\}$. Now we construct the order parameters for the $S_3$ SSB phase. The first one is a $\mathbb{Z}_2$ order parameter $\mathcal{O}_B$ given by the bulk anyon line $B$ stretches across the sandwich. From Eq. (7), Eq. (12), and Eq. (13), we find the order parameter $\mathcal{O}_B$ is odd under $s$ transformation:

$$U_s \cdot \mathcal{O}_B = \frac{\Psi_{sB}^{S_3}}{\sqrt{S_{AB}}} \mathcal{O}_B = (-1)\mathcal{O}_B, \tag{14}$$

and even under $r$ transformation. The other order parameter has two component and is fully faithful under $S_3$. Recall that there are two condensation channels for the $C$ anyon line. Let $\mathcal{O}^1$ and $\mathcal{O}^2$ be the local operators corresponding to the anyon line $C$ with condensation channel 1 and 2 on the reference boundary, respectively (and we fix the condensation channel on the right boundary). Using Eq. (7), Eq. (12), and Eq. (13), we have

$$U_s \cdot \mathcal{O}^1 = \mathcal{O}^1, \quad U_s \cdot \mathcal{O}^2 = (-1)\mathcal{O}^2 \tag{15}$$

and also

$$U_r \cdot \mathcal{O}^1 = \cos\left(\frac{2\pi}{3}\right)\mathcal{O}^1 + \sin\left(\frac{2\pi}{3}\right)\mathcal{O}^2, \quad U_r \cdot \mathcal{O}^2 = \sin\left(\frac{2\pi}{3}\right)\mathcal{O}^1 + \cos\left(\frac{2\pi}{3}\right)\mathcal{O}^2, \tag{16}$$

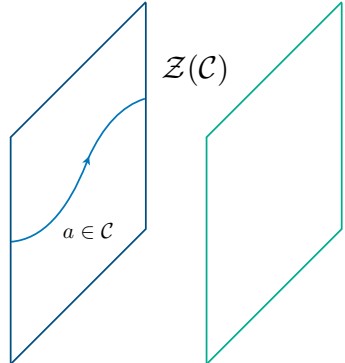

FIG. 3. Topological defect lines on the left boundary implement the fusion category $\mathcal{C}$ symmetry.

We then define a two-components order parameter

$$\boldsymbol{\mathcal{O}}_C = \begin{pmatrix} \mathcal{O}^1 + \mathcal{O}^2 \\ \mathcal{O}^1 - \mathcal{O}^2 \end{pmatrix}, \tag{17}$$

which transform under the $S_3$ generators as

$$U_s \cdot \boldsymbol{\mathcal{O}}_C = \begin{pmatrix} 0 & 1 \\ 1 & 0 \end{pmatrix} \boldsymbol{\mathcal{O}}_C \tag{18}$$

$$U_r \cdot \boldsymbol{\mathcal{O}}_C = \begin{pmatrix} \cos\left(\frac{2\pi}{3}\right) & \sin\left(\frac{2\pi}{3}\right) \\ \sin\left(\frac{2\pi}{3}\right) & \cos\left(\frac{2\pi}{3}\right) \end{pmatrix} \boldsymbol{\mathcal{O}}_C. \tag{19}$$

When the bulk is given by $\mathcal{Z}(\mathcal{B}) = \mathcal{B} \boxtimes \bar{\mathcal{B}}$ and the left boundary is the canonical gapped boundary (6), local operators are labeled by $\alpha = (a, a)$ with $a \in \mathcal{B}$, and the defect lines are labeled by $m \in \mathcal{B}$, but lifted into $(m, 1)$ in the bulk. The half-braiding matrix is given by [53]

$$\Psi_{m\alpha} = \frac{S_{(a,a),(m,1)}}{\sqrt{S_{(1,1),(a,a)}}} = S_{am}. \tag{20}$$

The symmetry action of the fusion category symmetry $\mathcal{C}$ on the local operator is then given by

$$U_m \cdot \mathcal{O}_\alpha = \frac{S_{ma}}{S_{1a}} \mathcal{O}_\alpha. \tag{21}$$

The result can be understood as follows: first the boundary line $m$ can be lifted to a simple anyon line $(m, 1)$ in the bulk, and the symmetry transformation is simply given by the mutual braiding between the bulk anyons $\alpha = (a, a)$ and $m$, which takes the same form as Eq.(21). We note that the a symmetry operator acts non-trivially on the local operator $\mathcal{O}_\alpha$ only when the corresponding bulk anyon $\mu$ is mapped to an confined anyon $m \in \mathcal{C}$ on the left boundary since the mutual braiding statistics between the set of condensed anyons is trivial.

A defect operator is an operator sitting at the end point of a topological defect line. This is represented in the sandwich construction as a bulk anyon line $\beta \in \mathcal{Z}(\mathcal{C})$ stretched across the bulk and connecting to a topological line $a \in \mathcal{C}$ on the left boundary as shown in Fig. 4(b). The action of the symmetries on a defect operator can be formulated in terms of a *lasso* diagram, which is shown on the left boundary in Fig. 4(b). When we have abelian anyons, this action can also be lifted and represented as a mutual braiding between the bulk anyon $\mu$ and $\beta$.

We note that there exist a set of symmetry operators that act non-trivially only on the defect operators, and act trivially on all the local operators. This set of symmetry operators corresponds to the set of condensed anyons on the left boundary. We can call the symmetry generated by such condensed anyon lines as a *dual symmetry*. As emphasized by Ref. [23, 25, 29, 34–36], we can think of the bulk anyon lines generate a *symmetry topological order* (or *categorical symmetry*), described by the MTC $\mathcal{Z}(\mathcal{C})$.

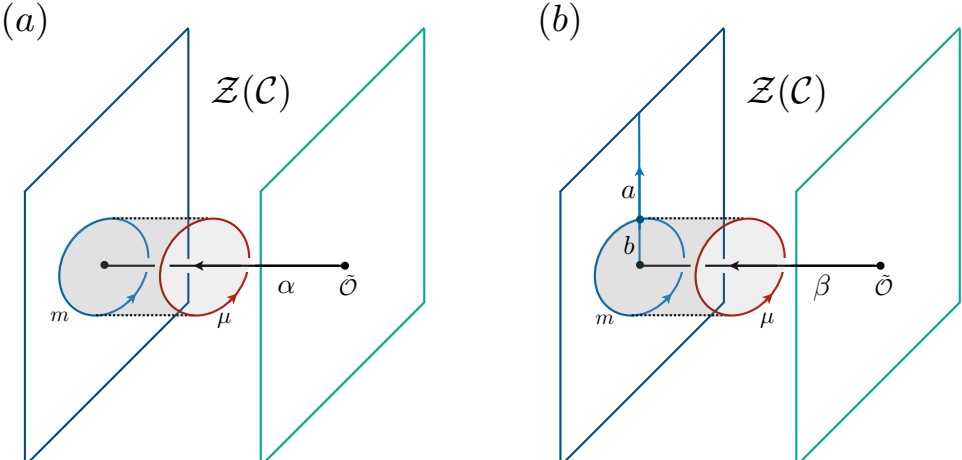

FIG. 4. Symmetry actions on the local and defect operators. (a) A local operator $\mathcal{O}_\alpha$ corresponds to a bulk anyon line $\alpha$ that condenses on the left topological boundary, and stretches across the sandwich. The symmetry $\mathcal{C}$ acts on the local operator as the mutual braiding between the bulk anyon line $\alpha$ and the boundary topological line $m \in \mathcal{C}$. The boundary line $m$ can be lifted to a bulk line $\mu$ and the symmetry transformation is given by the mutual braiding between the bulk anyons $\alpha$ and $\mu$. (b) A defect operator corresponds to a bulk anyon line $\beta$ stretched across the bulk and connecting to a topological line $a \in \mathcal{C}$ on the left boundary. The action of the symmetries on a defect operator can be formulated in terms of a *lasso* diagram (shown on the left boundary), which can also be lifted and became a mutual braiding between the bulk anyon $\mu$ and $\beta$.

## C. Dualities

Topological holography also provides a way to understand dualities between $(1+1)d$ theories. Given a bulk topological order $\mathcal{T}$, in general there can be anyonic symmetry that permutes the bulk anyons:

$$\rho : \alpha \to \rho(\alpha). \tag{22}$$

Each of this anyonic symmetry corresponds to an invertible codimension-1 twist defect $\mathcal{D}$ in the bulk, such that a bulk anyon is permuted when it passing through the defect. In the sandwich construction, we can insert this twist defect parallel to the sandwich, giving rise to a new $(1+1)d$ phase. If the twist defect satisfies a $\mathbb{Z}_2$ fusion rule, each of the twist defect of this kind corresponds to a duality between two different $(1+1)$D gapped phases. More specifically, consider two gapped topological gapped boundary conditions $|\mathcal{L}_1\rangle$ and $|\mathcal{L}_2\rangle$ on the right boundary. The corresponding partition functions are given by $Z_1 = \langle \mathcal{A}|\mathcal{L}_1\rangle$ and $Z_2 = \langle \mathcal{A}|\mathcal{L}_2\rangle$, where $\langle \mathcal{A}|$ is the left topological boundary. We say that there is a duality between these two gapped phases, if there exist a twist defect $\mathcal{D}$, such that $\mathcal{D}|\mathcal{L}_1\rangle = |\mathcal{L}_2\rangle$. Written in terms of the partition functions, this means that $Z_1' = \langle \mathcal{A}|\mathcal{D}|\mathcal{L}_1\rangle = Z_2$.

Suppose we tune the right boundary to the transition point between the gapped boundaries $|\mathcal{L}_1\rangle$ and $|\mathcal{L}_2\rangle$. Let's denote the corresponding nontopological boundary condition by $|\Psi\rangle$. The duality implies that the non-topological boundary condition has an important property: $\mathcal{D}|\Psi\rangle = |\Psi\rangle$. This means that the duality becomes a self-duality at the phase transition point as one can see explicitly from the invariance of the partition function under the insertion of the twist defect: $Z' = \langle \mathcal{A}|\mathcal{D}|\Psi\rangle = \langle \mathcal{A}|\Psi\rangle = Z$. Since we can move the twist defect to the left and fuse with the left topological boundary, the self-dual theory will satisfy

$$\begin{aligned} Z' &= \langle \mathcal{A}|\mathcal{D}|\Psi\rangle \\ &= \langle \mathcal{A}'|\Psi\rangle. \\ &= Z \end{aligned} \tag{23}$$

We will encounter examples in $\mathbb{Z}_2 \times \mathbb{Z}_2$ gauge theory where there exist a twist defect satisfies a $S_3$ fusion rule. This kind of twist defects give rise to a triality for the corresponding $(1+1)d$ systems. For two critical

boundaries $\Psi_1$ and $\Psi_2$ related by the twist defect of this kind: $\mathcal{D}|\Psi_1\rangle = |\Psi_2\rangle$, there is a relation between the partition functions of these two critical theories:

$$
\begin{aligned}
Z_2 &= \langle \mathcal{A}|\Psi_2\rangle \\
&= \langle \mathcal{A}|\mathcal{D}|\Psi_1\rangle \\
&= \langle \mathcal{A}'|\Psi_1\rangle,
\end{aligned}
\tag{24}
$$

where $\langle \mathcal{A}'| = \langle \mathcal{A}|\mathcal{D}$. This shows that the partition function of theory 2 is given by the sandwich of theory 1 but with a different choice of the left topological boundary. We can thus use this relation to write down the partition function for theory 2 by using the characters of theory 1.

More generally, for any two topological gapped boundaries $\mathcal{A}$ and $\mathcal{B}$ not related by an anyonic symmetry, there is a corresponding duality between the two quantum systems. This class of dualities is more general since the fusion category symmetries corresponding to the gapped boundaries $\mathcal{A}$ and $\mathcal{B}$ might not be equivalent in the sense of fusion categories. Nevertheless, this is still a duality between the two quantum systems as there is still a one-to-one mapping between the local operators. A conjecture made by Ref. [23, 35] is that two quantum systems with fusion category symmetries $\mathcal{C}$ and $\mathcal{C}'$ are dual to each other if $\mathcal{Z}(\mathcal{C}) \cong \mathcal{Z}(\mathcal{C}')$. An explicit example we will see below is that a system with a $\mathbb{Z}_2 \times \mathbb{Z}_2$ symmetry and a mixed anomaly is dual to a system with a $\mathbb{Z}_4$ symmetry.

### D.  Rational CFTs

Here we give a brief review of RCFTs and explain the relation to the sandwich picture. Generally, a $(1+1)d$ CFT has both a chiral vertex algebra $\mathcal{V}_L$, and the anti-chiral one $\mathcal{V}_R$. Below we assume that they are isomorphic, so we can just focus on $\mathcal{V}_L$. The Hilbert space of the CFT on a circle decomposes as

$$
\mathcal{H} = \bigoplus_{a,b} M_{ab}\mathcal{H}_a \otimes \overline{\mathcal{H}}_b,
\tag{25}
$$

Here $\mathcal{H}_a$'s are irreducible representations of the chiral algebra $\mathcal{V}_L$, and $M_{ab}$ is a modular invariant matrix with non-negative entries. This data also enters the untwisted torus partition function:

$$
\begin{aligned}
Z_{00}(\tau,\bar{\tau}) &= \mathrm{Tr}_{\mathcal{H}} e^{-\beta H}, \\
&= \sum_{a,b} M_{ab}\chi_a(\tau)\bar{\chi}_b(\bar{\tau}),
\end{aligned}
\tag{26}
$$

where $\chi_a(\tau)$ is the character of the irreducible representation of the chiral algebra, defined as

$$
\chi_a(\tau) = \mathrm{Tr}_{H_a} e^{2\pi i \tau (L_0 - \frac{c}{24})}.
\tag{27}
$$

Here $\tau$ is the modular parameter of the torus, and $L_n$ is the $n$-th Virasoro generator. The diagonal RCFTs correspond to choosing $M_{ab} = \delta_{ab}$.

There is a well-established bulk-boundary correspondence between $(1+1)d$ RCFTs and $(2+1)d$ chiral topological orders on a strip[42–45]. A chiral algebra uniquely determines a modular tensor category $\mathcal{B} = \mathrm{Rep}(\mathcal{V}_L)$, whose simple objects are in one-to-one correspondence with the irreducible representations of $\mathcal{V}_L$. The bulk chiral topological order is described by the MTC $\mathcal{B}$. The edge modes are described by the corresponding chiral CFT. When viewed as a quasi-one-dimensional system, the left and right edges of the strip together describe a full non-chiral RCFT. In general, for non-diagonal RCFT, there is a gapped domain wall inserted at the middle of the strip, and the diagonal theory corresponds to inserting a trivial domain wall (no domain wall). The sandwich picture discussed in the previous section is obtained by folding along the domain wall so that it becomes a gapped boundary of a double topological order $\mathcal{Z}(\mathcal{B}) = \mathcal{B} \boxtimes \overline{\mathcal{B}}$. For diagonal theories, the topological defect lines on the gapped boundary is simply described by the fusion category $\mathcal{B}$.

We can think of the partition function Eq. (26) as the inner product $\langle \mathcal{A}|\Psi\rangle$ as follows. We denote the bulk anyons as $(a,b) \in \mathcal{B} \boxtimes \overline{\mathcal{B}}$. Define a state corresponding to the gapped boundary $\mathcal{A}_\mathcal{S}$:

$$
|\mathcal{A}\rangle = \sum_{a,b} w_{(a,b)}|(a,b)\rangle = \sum_{a,b} M_{a,b}|(a,b)\rangle,
\tag{28}
$$

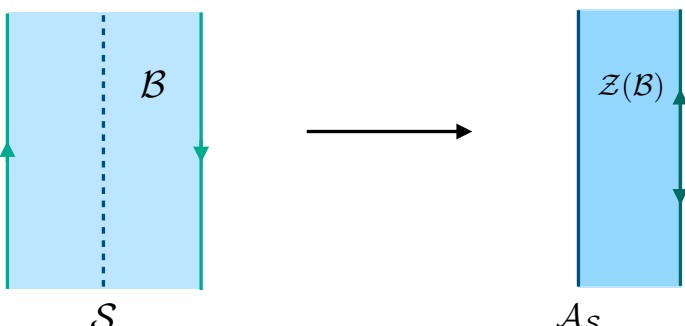

FIG. 5. A $(1+1)d$ RCFTs corresponds to a $(2+1)d$ chiral topological orders $\mathcal{B}$ on a strip with an insertion of a gapped domain wall $\mathcal{S}$ in the middle. The sandwich picture is obtained by folding along the gapped domain wall, which results in a double topological order $\mathcal{Z}(\mathcal{B}) = \mathcal{B} \boxtimes \overline{\mathcal{B}}$ in the bulk with a gapped boundary condition $\mathcal{A}_{\mathcal{S}}$.

where $M_{a,b}$ is the modular invariant matrix. And define a state corresponding to the CFT boundary condition:

$$|\Psi\rangle = \sum_{a,b} \chi_a(\tau)\bar{\chi}_b(\bar{\tau})|(a,b)\rangle. \tag{29}$$

By taking the inner product $\langle \mathcal{A}|\Psi\rangle$, we recover the partition function Eq. (26).

The bulk of the sandwich is not unique. Consider a topological order $\mathcal{Z}(\mathcal{C})$ on a solid torus with an insertion of an anyon $\alpha$ and the boundary being a RCFT. The partition function of the system is given by $Z_\alpha$. Suppose there is another MTC $\mathcal{Z}(\mathcal{C}')$ which has a gapped domain wall with $\mathcal{Z}(\mathcal{C})$. Now we create a pair of gapped domain walls $I$ and $I^{-1}$ such that the bulk in between $I$ and $I^{-1}$ is the MTC $\mathcal{Z}(\mathcal{C}')$. We then move the domain wall $I$ along the non-contractible $S^1$ direction and annihilate with the domain wall $I^{-1}$ from the other side. The bulk is now given by the new topological order $\mathcal{Z}(\mathcal{C}')$. The original anyon $\alpha$ inserted in the middle in general becomes some linear combination of the anyons in $\mathcal{Z}(\mathcal{C}')$:

$$\alpha = \sum_{\mu' \in \mathcal{Z}(\mathcal{C}')} B_{\alpha,\mu'}\mu', \tag{30}$$

where each entry in $B_{\alpha,\mu'}$ is a non-negative integer, which is sometimes called a branching or a tunneling matrix[54–56]. This implies the following relations between the partition functions:

$$Z_\alpha = \sum_{\mu' \in \mathcal{Z}(\mathcal{C}')} B_{\alpha,\mu'}Z_{\mu'}, \tag{31}$$

where $Z_{\mu'}$ is the torus partition functions of the topological order $\mathcal{Z}(\mathcal{C}')$ with the insertion of anyon $\mu'$ and the boundary being the same RCFT. Inserting Eq. (31) into the RCFT partition function Eq. (4) in the sandwich picture, we have

$$
\begin{aligned}
Z &= \sum_{\mu' \in \mathcal{Z}(\mathcal{C}')} \left( \sum_{\alpha \in \mathcal{T}} w_\alpha B_{\alpha,\mu'} \right) Z_{\mu'} \\
&= \sum_{\mu' \in \mathcal{Z}(\mathcal{C}')} w_{\mu'} Z_{\mu'}.
\end{aligned} \tag{32}
$$

A simple example is given by taking $\mathcal{Z}(\mathcal{C})$ to be the $\mathbb{Z}_2$ toric code and $\mathcal{Z}(\mathcal{C}')$ to be the double Ising theory. Anyons in the toric code are mapped to the anyons in the double Ising theory following the rules of anyon

condensation:

$$1 = 1 \oplus \psi\bar{\psi},$$
$$e = \sigma\bar{\sigma},$$
$$m = \sigma\bar{\sigma},$$
$$f = \psi \oplus \bar{\psi}. \tag{33}$$

Applying Eq. (31), we obtain

$$Z_1 = |\chi_1|^2 + |\chi_\psi|^2,$$
$$Z_e = |\chi_\sigma|^2,$$
$$Z_m = |\chi_\sigma|^2,$$
$$Z_f = \chi_1\overline{\chi}_\psi + \chi_\psi\overline{\chi_1}. \tag{34}$$

The torus partition function of the Ising CFT in the $\mathbb{Z}_2$ toric code basis is given by $Z_{\text{Ising}} = Z_1 + Z_e$ according to Eq. (4), where we have chosen the $e$-condensed boundary condition on the left boundary. By using Eq. (32) and Eq. (34), we can write it in terms of the original characters in the Ising theory as

$$Z_{\text{Ising}} = Z_1 + Z_e = |\chi_1|^2 + |\chi_\psi|^2 + |\chi_\sigma|^2. \tag{35}$$

The Kramers–Wannier duality corresponds to the electro-magnetic (EM) twist defect $\mathcal{D}_{\text{EM}}$ in the $\mathbb{Z}_2$ toric code. By inserting the EM twist defect $\mathcal{D}_{\text{EM}}$ and fusing with the left boundary, the left boundary condition is changed to an $m$-condensed boundary. In the Ising CFT, the Kramers–Wannier duality becomes a self-duality, which can be understood as a statement that the dual partition function is the same as the original partition function. We can check this explicitly in the sandwich picture as follows:

$$Z'_{\text{Ising}} = \langle \mathcal{A}_e | \mathcal{D}_{\text{EM}} | \Psi \rangle \tag{36}$$
$$= \langle \mathcal{A}_m | \Psi \rangle. \tag{37}$$
$$= Z_1 + Z_m \tag{38}$$
$$= |\chi_1|^2 + |\chi_\psi|^2 + |\chi_\sigma|^2, \tag{39}$$

which is the same as the original partition function of the Ising CFT Eq. (35).

### 1. Twisted partition function

For a theory with $G$ symmetry, one can define twisted partition functions, namely partition functions with in the presence of a background $G$ gauge field. In $(1+1)d$ and when the spacetime manifold is a torus, the gauge equivalence classes of background gauge field are labeled by a pair of commuting group elements $g, h \in G$ (up to conjugation) corresponding to twists in the spatial and temporal directions. Without loss of generality, we may assume $G$ is Abelian and has no anomaly, so the twisted partition function has a well-defined value.

We now derive a general expression of the twisted partition function of RCFTs by using the sandwich picture. We consider an RCFT with an abelian anomaly-free $G$ symmetry. Let $Z_{g,h}$ be the twisted partition function of the RCFT with an insertion of $h$ defect at a fixed time and a $g$ twisted boundary condition. Let $W^t_{m_g}, W^x_{m_h} \in \mathcal{C} \cong \text{Vec}_G$ denote the topological defect lines on the reference boundary that generate $g$ and $h$ symmetry transformations, respectively. Also let $W^t_{\mu_g}, W^x_{\nu_h} \in \mathcal{Z}(\mathcal{C}) \cong D(G)$ be the corresponding anyon lines in the bulk, which could be non-simple lines. The twisted partition function is given by

$$Z_{g,h} = \langle \mathcal{A} | W^t_{\mu_g} W^x_{\nu_h} | \psi \rangle. \tag{40}$$

To proceed, we shrink the radius of the left boundary so that it becomes an Lagrangian algebra anyon $\mathcal{A}$. Inserting an topological defect line $W^t_{m_g}$ corresponds to fusing the corresponding bulk anyon $\mu_g$ to the

Lagrangian algebra anyon $\mathcal{A}$:

$$\mu_g \times \mathcal{A} = \bigoplus_{\alpha \in \mathcal{Z}(\mathcal{C})} w_\alpha \mu_g \times \alpha \tag{41}$$

$$= \bigoplus_{\alpha, \gamma \in \mathcal{Z}(\mathcal{C})} w_\alpha N^\gamma_{\mu_g, \alpha} \gamma. \tag{42}$$

We then have a corresponding state:

$$|W^t_{\mu_g} \mathcal{A}\rangle := W^t_{\mu_g} |\mathcal{A}\rangle = \sum_{\alpha, \gamma \in \mathcal{Z}(\mathcal{C})} w_\alpha N^\gamma_{\mu_g, \alpha} |\gamma\rangle. \tag{43}$$

The twisted partition function becomes

$$Z_{g,h} = \langle W^t_{\mu_g} \mathcal{A} | W^x_{\nu_h} | \psi \rangle. \tag{44}$$

Note that $W^x_{\nu_h}$ is an anyon line wrapping around the composite anyon $\mu_g \times \mathcal{A}$ at the center, which gives a mutual braiding phase between $\nu_h$ and $\mu_g \times \mathcal{A}$. With this observation, we finally arrive at the general expression of the twisted partition function:

$$Z_{g,h} = \langle W^t_{\mu_g} \mathcal{A} | W^x_{\nu_h} | \psi \rangle \tag{45}$$

$$= \sum_{\alpha, \gamma \in \mathcal{Z}(\mathcal{C})} w_\alpha N^\gamma_{\mu_g, \alpha} \frac{S_{\gamma \nu_h}}{S_{0\gamma}} Z_\gamma. \tag{46}$$

For example, in the Ising CFT, the $\mathbb{Z}_2$ symmetry is generated by the $m$ anyon line. The twisted partition function $Z_{01}$ is given by

$$Z_{0,1} = \sum_{\alpha \in A_e} \frac{S_{\alpha m}}{S_{1\alpha}} Z_\alpha \tag{47}$$

$$= Z_1 - Z_e \tag{48}$$

$$= |\chi_1|^2 + |\chi_\psi|^2 - |\chi_\sigma|^2, \tag{49}$$

where $A_e = \{1, e\}$ is the set of anyons of the $e$-condensed boundary. The twisted partition function $Z_{1,0}$ is

$$Z_{1,0} = \sum_{\alpha \in m \times A_e} Z_\alpha \tag{50}$$

$$= Z_m + Z_f \tag{51}$$

$$= |\chi_\sigma|^2 + \chi_1 \overline{\chi_\psi} + \chi_\psi \overline{\chi_1}. \tag{52}$$

Finally, the twisted partition function $Z_{1,1}$ is

$$Z_{1,1} = \sum_{\alpha \in m \times A_e} \frac{S_{\alpha m}}{S_{1\alpha}} Z_\alpha \tag{53}$$

$$= Z_m - Z_f \tag{54}$$

$$= |\chi_\sigma|^2 - \chi_1 \overline{\chi_\psi} - \chi_\psi \overline{\chi_1}. \tag{55}$$

## III.   GAPPED PHASES

In this section we study gapped phases from the perspective of topological holography (similar results were presented in Ref. [46, 47, 57]). We assume that the bulk theory is a Drinfeld center $\mathcal{Z}(\mathcal{C})$, and the reference boundary is the canonical one. The sandwich models a $(1+1)d$ system with fusion category $\mathcal{C}$. Suppose the corresponding Lagrangian algebra is $\mathcal{A}_0$. Now we can choose the right boundary condition, which can

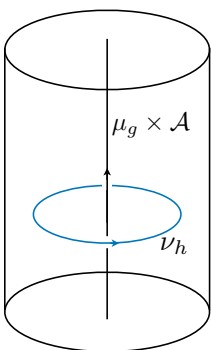

FIG. 6. The twisted partition function $Z_{gh}$ of an RCFT in the anyon basis is equivalent to the partition function of the $(2+1)$D topological order on an solid torus with an insertion of an composite anyon $\mu_g \times \mathcal{A}$ along the time-direction, which is wrapped around by another anyon $\nu_h$ in the spatial direction.

be described in two different but equivalent ways: one can either choose a module category $\mathcal{M}$ over $\mathcal{C}$ (the canonical reference boundary has $\mathcal{C}$ itself as the module category), or a Lagrangian algebra $\mathcal{A}$ in $\mathcal{Z}(\mathcal{C})$.

Once the right boundary condition is given, we have a $(1+1)d$ gapped system. We would like to emphasize that it is ambiguous to associate a topological gapped boundary condition to a $(1+1)d$ gapped phase without specifying the boundary condition on the reference boundary of the sandwich. To understand the physics, first of all we need to determine the local operators, which are generated by Wilson lines that can end on both boundaries. If there are nontrivial local operators (other than the identity Wilson line), then the $\mathcal{C}$ fusion category symmetry is broken (partially or completely, depending on the right boundary condition). As an example, if the right boundary condition is also the canonical one, then the $\mathcal{C}$ fusion category symmetry is completely broken. We can then classify gapped states according to the eigenvalues of these Wilson lines. Alternatively, one can imagine inserting different Wilson lines parallel to the boundary in the bulk of the sandwich.

When no such Wilson lines exist, the full $\mathcal{C}$ symmetry is preserved. In this case we say that this is a completely symmetric phase. When this happens, the corresponding module category must have a unique object, i.e. it has a fiber functor. It is possible that there are multiple fiber functors, corresponding to distinct fusion category symmetry protected phases.

In general, a part of the $\mathcal{C}$ fusion category symmetry is preserved. Physically, this is the subcategory $\tilde{\mathcal{C}}$ of boundary topological lines whose actions on the local Wilson lines (given by half braiding) are trivial.

To summarize, a gapped state in the sandwich construction comes naturally with the following datum:

1. The right boundary condition, given by a Lagrangian algebra $\mathcal{A}$ (or a module category over $\mathcal{C}$).

2. The algebra of local Wilson lines $\{W_x\}$, which is determined by the right boundary condition.

3. The unbroken fusion category symmetry $\tilde{\mathcal{C}}$. Again, this is fully determined by the right boundary condition.

4. The eigenvalues of the local Wilson lines.

We will thus label a gapped phase by $(\mathcal{A}, \tilde{\mathcal{C}}, \{W_x\})$ where $W_x$ are the eigenvalues of the local Wilson lines.

To illustrate the main idea, here we provide some examples on topological holographic description of gapped phases, which will be relevant for later parts of the work.

### A. Phases with modular category symmetry

First, let us assume that $\mathcal{C}$ is a MTC, so $\mathcal{Z}(\mathcal{C}) = \mathcal{C} \boxtimes \bar{\mathcal{C}}$. The canonical boundary is given by

$$\mathcal{A}_0 = \sum_{a \in \mathcal{C}} (a, a). \tag{56}$$

The TDLs are labeled by $a \in \mathcal{C}$, and the fusion category is precisely $\mathcal{C}$ (forgetting the braiding).

A class of right boundary condition is given by the following Lagrangian algebra:

$$\mathcal{A}_\varphi = \sum_{a \in \mathcal{C}} (a, \varphi(a)). \tag{57}$$

Here $\varphi \in \mathrm{Aut}(\mathcal{C})$ is a topological symmetry of $\mathcal{C}$. Then the local operators are generated by Wilson lines of the form $(a, a)$ where $a = \varphi(a)$. If $\varphi = 1$, then one can see that the $\mathcal{C}$ fusion category symmetry is completely broken. Otherwise there is a subcategory of $\mathcal{C}$ with trivial braiding with $\varphi$-invariant anyons that remains unbroken.

Intuitively, one can view the sandwich simply as a cylinder of $\mathcal{C}$, where the right boundary becomes an invertible defect line $\varphi$. The number of ground states on the cylinder is given by the number of $\varphi$-invariant anyons. The different states can be obtained by inserting anyon lines parallel to the defect line. In the simplest case when $\varphi = 1$, the $\mathcal{C}$ symmetry is completely broken, and ground states correspond to anyon.

## B. Gapped phases with $G$ symmetry

For a discrete gauge theory, the gapped boundary is labeled by a pair $(H, \omega)$ (up to conjugacy), where $H \subset G$ is a subgroup of $G$ and $[\omega] \in \mathcal{H}^2[H, \mathrm{U}(1)]$. Mathematically, the pair $(H, \omega)$ classifies module categories over the fusion category $\mathrm{Vec}_G$. Alternatively, one can also characterize gapped boundaries by their Lagrangian algebras.

In the sandwich, we always choose the the reference boundary to be the one that completely breaks $G$ symmetry, i.e. the one labeled by $(\{1\}, 0)$. This corresponds to the pure charge condensation boundary in the $G$ gauge theory. The topological defect lines corresponding to the confined fluxes on the reference boundary are labeled by group elements in $G$, which generate the $G$ symmetry. Mathematically, they form the $\mathrm{Vec}_G$ category. The sandwich construction realizes different gapped phases in $(1+1)d$ with different choices of the gapped boundary on the right. Since the gapped boundaries on the right are classified by the pair $(H, \omega)$, we conclude that the classification of the gapped phases in $(1+1)d$ is also given by the same pair. A complete spontaneous symmetry breaking phase corresponds to choosing both boundaries to be the charge-condensed boundary $(\{1\}, 0)$. The order parameters are the charge lines that stretch across the sandwich and condense on both boundaries. More generally, $H$ represents the unbroken subgroup of $G$ and $\omega$ labels a $(1+1)d$ SPT phase under the $H$ symmetry. We discuss various simple examples below.

### 1. Gapped phases with $G = \mathbb{Z}_2$ symmetry

We begin with the discussion on gapped phases with $G = \mathbb{Z}_2$ symmetry [27]. The bulk of the sandwich is a $\mathbb{Z}_2$ gauge theory. On the reference boundary, we choose $e$ condensed boundary. If we also choose $e$ condensed boundary on the other side of the sandwich, this becomes a $\mathbb{Z}_2$ SSB state. To see that, we consider the $\mathbb{Z}_2$ order parameter $\mathcal{O}_{\mathbb{Z}_2}$, which is represented as a $e$ line stretched between the two boundaries. The $\mathbb{Z}_2$ symmetry is implemented by the $m$ line on the reference boundary. Due to the non-trivial braiding statistics between $e$ and $m$, we have an order parameter which transforms non-trivially under the symmetry. Fixing the state on the reference boundary, we still have two degenerate ground states related by threading $m$ line on the right boundary. We denote the two states by $|1\rangle$ and $|m\rangle$. By the mutual braiding between $e$ and $m$, the order parameter has opposite expectation value for these two state:

$$\langle 1 | \mathcal{O}_{\mathbb{Z}_2} | 1 \rangle = \langle 1 | W_e | 1 \rangle = 1, \tag{58}$$

$$\langle m | \mathcal{O}_{\mathbb{Z}_2} | m \rangle = \langle m | W_e | m \rangle = -1, \tag{59}$$

which is the non-trivial signature of the SSB phase.

Now we discuss $m$ condensed boundary. Naively, it seems that there are still two degenerate ground states related by threading a $e$ line along the boundary. However, the $e$ line can be adiabatically moved to the reference boundary and be absorbed. We actually have a unique ground state with $\mathbb{Z}_2$ symmetry, hence it is a $\mathbb{Z}_2$ symmetric trivial state. Relatedly, there are no nontrivial local order parameter in this case, since no anyons can terminate on both ends.

The EM exchange defect in the $\mathbb{Z}_2$ gauge theory corresponds to the Kramers-Wannier duality between the $\mathbb{Z}_2$ SSB and $\mathbb{Z}_2$ symmetric phases as it exchanges the $e$ and $m$ condensed boundaries.

### 2. $G = S_3$

Let us consider a non-Abelian example with $G = S_3 = \{r, s | r^3 = s^2 = 1, srs = r^{-1}\}$. The bulk is the $S_3$ gauge theory with 8 anyon types. Following the convention in [58], we label them as $A, B, \ldots, H$. $A, B$ and $C$ are the three gauge charges: $A$ corresponds to the identity representation, $B$ is the one-dimensional representation and $C$ is the two-dimensional representation. $D$ and $E$ are the two fluxes with conjugacy class $[s] = \{s, sr, sr^2\}$. $F, G, H$ are the three fluxes with conjugacy class $[r] = \{r, r^2\}$. Note that the pure fluxes $D$ and $F$ are bosons, while the "dyons" $E, G, H$ are not.

The reference boundary is characterized by the pure charge condensation $A + B + 2C$. The right boundary has four possibilities:

1. $A + B + 2C$ and $H = \{1\}$. In this case, the $S_3$ symmetry is completely broken, and there are 6 ground states.

2. $A + D + F$ and $H = S_3$. There are no Wilson lines connecting the two boundaries, so the $S_3$ symmetry is unbroken. This is the fully symmetric state.

3. $A + B + 2F$ and $H = \mathbb{Z}_3$. The only nontrivial local operator is the local $B$ Wilson line, which braids trivially with the $r$ and $r^2$ lines on the reference boundary. Thus the remaining symmetry is $\mathbb{Z}_3$. There are 2 ground states as expected from the $S_3 \to \mathbb{Z}_3$ SSB.

4. $A + C + D$ and $H = \mathbb{Z}_2$. The nontrivial local operators are given by $\mathcal{O}_C^1, \mathcal{O}_C^2$, where 1 and 2 label the condensation channel on the reference boundary (the channel is unique for $C$ on the right). They form an irreducible representation under the $S_3$ symmetry, so the remaining global symmetry is trivial. However, there are three ground states. This is in fact what we expect for the $S_3 \to \mathbb{Z}_2$ SSB: since $\mathbb{Z}_2$ is not a normal subgroup, each (short-range-correlated) symmetry-breaking state corresponds to a different $\mathbb{Z}_2$ subgroup. More formally, the three ground states form a reducible representation, decomposing into the identity rep and the 2-dimensional rep.

### 3. Gapped phases with $G = \mathbb{Z}_4$ symmetry

For quantum systems with $G = \mathbb{Z}_4$ symmetry, we choose the bulk to be a $\mathbb{Z}_4$ gauge theory. There are three gapped boundaries with the following Lagrangian algebra of condensed anyons

$$\mathcal{L}_0 = 1 \oplus e \oplus e^2 \oplus e^3, \tag{60}$$

$$\mathcal{L}_1 = 1 \oplus e^2 \oplus m^2 \oplus e^2 m^2, \tag{61}$$

$$\mathcal{L}_2 = 1 \oplus m \oplus m^2 \oplus m^3. \tag{62}$$

The reference boundary is a $e$ condensed boundary $\mathcal{L}_0$. Choosing the right boundary to be $\mathcal{L}_0$, $\mathcal{L}_1$, and $\mathcal{L}_2$ corresponds to the $\mathbb{Z}_4$ SSB, partially $\mathbb{Z}_2$ ordered, and $\mathbb{Z}_4$ symmetric phases, respectively. The order parameter of the $\mathbb{Z}_4$ SSB phase is given by the $e$ line connecting the two boundaries, and, for the partially $\mathbb{Z}_2$ ordered phase, the order parameter is given by the $e^2$ line. Due to the EM exchange automorphism in the $\mathbb{Z}_4$ gauge theory, which exchange $\mathcal{L}_0$ and $\mathcal{L}_2$ boundaries, we have a generalized Kramers-Wannier duality between the $\mathbb{Z}_4$ SSB phase and the $\mathbb{Z}_4$ symmetric phase, and the partially $\mathcal{Z}_2$ ordered phase is invariant under the duality.

### 4. Gapped phases with $G = \mathbb{Z}_2 \times \mathbb{Z}_2$ symmetry

We move on to the examples with $G = \mathbb{Z}_2 \times \mathbb{Z}_2$ symmetry. The bulk of the sandwich is a $\mathbb{Z}_2 \times \mathbb{Z}_2$ gauge theory. We denotes the first $\mathbb{Z}_2$ symmetry by $\mathbb{Z}_2^a$ and the second one by $\mathbb{Z}_2^b$. There are 6 gapped boundaries for the $\mathbb{Z}_2 \times \mathbb{Z}_2$ gauge theory as summarized in Table. I. We choose the $e$ condensed boundary $\mathcal{L}_1$ as our reference. Table. I show the gapped boundaries of the $\mathbb{Z}_2 \times \mathbb{Z}_2$ gauge theory.

Unlike all the previous examples, in this case we have two fully symmetric gapped boundaries, $\mathcal{L}_5$ and $\mathcal{L}_6$. Physically, we know that they correspond to the trivial symmetric phase and the $\mathbb{Z}_2^a \times \mathbb{Z}_2^b$ SPT phase. Let us see how to understand it from the sandwich construction. To show that the two states corresponding to

| Lagrangian algebra | Unbroken subgroup | $H^2(G, \mathrm{U}(1))$ | Order parameters |
|---|---|---|---|
| $\mathcal{L}_1 = 1 \oplus e_1 \oplus e_2 \oplus e_1 e_2$ | $1$ | $1$ | $\mathcal{O}_{\mathbb{Z}_2^a} = W_{e_1}, \, \mathcal{O}_{\mathbb{Z}_2^b} = W_{e_2}$ |
| $\mathcal{L}_2 = 1 \oplus m_1 \oplus e_2 \oplus m_1 e_2$ | $\mathbb{Z}_2^a$ | $1$ | $\mathcal{O}_{\mathbb{Z}_2^b} = W_{e_2}$ |
| $\mathcal{L}_3 = 1 \oplus e_1 \oplus m_2 \oplus e_1 m_2$ | $\mathbb{Z}_2^b$ | $1$ | $\mathcal{O}_{\mathbb{Z}_2^a} = W_{e_1}$ |
| $\mathcal{L}_4 = 1 \oplus e_1 e_2 \oplus m_1 m_2 \oplus f_1 f_2$ | $\mathbb{Z}_2^{\mathrm{diag}}$ | $1$ | $\mathcal{O}_{\mathbb{Z}_2^D} = W_{e_1 e_2}$ |
| $\mathcal{L}_5 = 1 \oplus e_1 m_2 \oplus m_1 e_2 \oplus f_1 f_2$ | $\mathbb{Z}_2^a \times \mathbb{Z}_2^b$ | $(-1)^{g_1 h_2}$ | - |
| $\mathcal{L}_6 = 1 \oplus m_1 \oplus m_2 \oplus m_1 m_2$ | $\mathbb{Z}_2^a \times \mathbb{Z}_2^b$ | $1$ | - |

TABLE I. Gapped boundaries of a $\mathbb{Z}_2 \times \mathbb{Z}_2$ gauge theory. The first column shows the Lagrangian algebra. The second and the third columns show the symmetry and the SPT index realized in the sandwich construction by choosing the reference boundary to be $\mathcal{L}_1$. The fourth column shows the order parameters as the Wilson line.

choosing $\mathcal{L}_5$ and $\mathcal{L}_6$ are distinct, we consider the junction between them. It is easy to see that now there emerges a local operator at the junction: a Wilson line of $e_1$ starting from the reference boundary, splitting into $m_2$ and $e_1 m_2$. Denote it by $O_1$. Similarly, we have a Wilson operator $O_2$ of a $e_2$ line splitting into $m_1$ and $e_2 m_1$. In particular, from the braiding we see that $O_1 O_2 = -O_2 O_1$. Therefore the junction must harbor a zero mode. Notice that $O_1$ can be deformed into a $m_2$ line along the strip together with a $e_1$ line ending on the reference boundary, so one can think of $O_1$ as the effective generator of the $\mathbb{Z}_2^b$ symmetry at the junction. This the familiar $\mathbb{Z}_2^a \times \mathbb{Z}_2^b$ projective rep. that appears at the edge of the SPT state. This shows that $\mathcal{L}_5$ and $\mathcal{L}_6$ correspond to distinct phases. It is a common convention to define $\mathcal{L}_5$ as the trivial phase, and then $\mathcal{L}_6$ is the nontrivial SPT phase.

The automorphism in the $\mathbb{Z}_2 \times \mathbb{Z}_2$ gauge theory is discussed in Ref. [59]. All automorphisms are generated by the EM exchange for the $\mathbb{Z}_2^a$ and $\mathbb{Z}_2^b$ gauge theory:

$$\sigma_i : e_i \leftrightarrow m_i, \tag{63}$$

and the twist defect from the gauged $\mathbb{Z}_2 \times \mathbb{Z}_2$ SPT:

$$s : e_i \to e_i, \; m_i \to m_i e_j, i \neq j. \tag{64}$$

Choosing the right boundaries to be $\mathcal{L}_i$ for $i = 1$ to $6$ realizes phases with the unbroken subgroup and the SPT index $H^2(G, \mathrm{U}(1))$ shown in the second and third columns in Table. I. The automorphisms correspond to the Kramers-Wannier duality between these gapped phases, which is discussed in Ref. [59]. The order parameters are given by the Wilson lines $W_\alpha$ connecting the two gapped boundaries, where $\alpha$ is in the common anyon set in the two Lagrangian algebra $\mathcal{L}_1$ and $\mathcal{L}_i$.

## IV. PHASE TRANSITIONS

### A. Phase transitions between SPT phases

In this section, we illustrate how to obtain the CFT partition function for a transition between different SPT phases by using topological holography and duality. This is an explicit application of Eq. (23). Ref. [59] contains somewhat similar discussion based on Fourier transformation of finite groups. Our analysis however is done entirely in the anyon basis, which is more general.

To illustrate the main idea, we consider systems with $G = \mathbb{Z}_2 \times \mathbb{Z}_2$ symmetry. Our starting point is the Ising$^2$ transition between the $\mathbb{Z}_2 \times \mathbb{Z}_2$ SSB and the trivial symmetric phase. This is realized by tuning the boundary of the sandwich to be at the phase transition point between the gapped boundaries $\mathcal{L}_1$ and $\mathcal{L}_6$. We denote the corresponding state on the right boundary by $|\psi_{1,6}\rangle$. The partition function of the Ising$^2$ CFT is given by

$$Z^{\mathrm{Ising}^2} = \langle \mathcal{L}_1 | \psi_{1,6} \rangle \tag{65}$$

$$= \sum_{\nu \in \mathcal{L}_1} \mathcal{Z}_\nu \tag{66}$$

$$= (|\chi_1|^2 + |\chi_\psi|^2)^2 + 2(|\chi_1|^2 + |\chi_\psi|^2)|\chi_\sigma|^2 + |\chi_\sigma|^4 \tag{67}$$

$$= (|\chi_1|^2 + |\chi_\psi|^2 + |\chi_\sigma|^2)^2, \tag{68}$$

where we have used $\mathcal{L}_1 = 1 \oplus e_1 \oplus e_2 \oplus e_1 e_2$ and Eq (34) in the second equality.

Knowing this CFT partition function allows us to obtain the partition function for other phase transitions related by the duality. In particular, we can write down the partition function for the transition between the $\mathbb{Z}_2 \times \mathbb{Z}_2$ SPT and the trivial symmetric phase. We tune the right boundary of the sandwich to be at the critical point between the gapped boundaries $\mathcal{L}_5$ and $\mathcal{L}_6$. We denote the corresponding state as $|\psi_{5,6}\rangle$ and the partition function of the SPT transition is $\mathcal{Z}_{\mathrm{SPT}} = \langle \mathcal{L}_1 | \psi_{5,6} \rangle$. Consider a twist defect $\mathcal{D} = \mathcal{D}_{\sigma_1 \sigma_2} \circ \mathcal{D}_s$. The twist defect $\mathcal{D}$ acts on the gapped boundaries as

$$\mathcal{D}|\mathcal{L}_1\rangle = |\mathcal{L}_6\rangle, \ \mathcal{D}|\mathcal{L}_6\rangle = |\mathcal{L}_5\rangle. \tag{69}$$

Therefore, we have

$$|\psi_{5,6}\rangle = \mathcal{D}|\psi_{1,6}\rangle. \tag{70}$$

The partition function for the SPT transition is then given by inserting the twist defect $\mathcal{D}$:

$$Z^{\mathrm{SPT}} = \langle \mathcal{L}_1 | \psi_{5,6} \rangle \tag{71}$$
$$= \langle \mathcal{L}_1 | \mathcal{D} | \psi_{1,6} \rangle \tag{72}$$
$$= \langle \mathcal{L}_5 | \psi_{1,6} \rangle \tag{73}$$
$$= \sum_{\nu \in \mathcal{L}_5} \mathcal{Z}_\nu \tag{74}$$
$$= (|\chi_1|^2 + |\chi_\psi|^2)^2 + 2|\chi_\sigma|^4 + (\chi_1 \overline{\chi}_\psi + \chi_\psi \overline{\chi_1})^2, \tag{75}$$

where, in the third equality, we have used the fact that $\langle \mathcal{L}_1 | \mathcal{D} = \langle \mathcal{L}_5 |$. Eq. (75) is exactly the partition function obtained in Ref. [60].

## B. Symmetry enriched quantum critical points

The topological holography can also be used to understand the symmetry enriched quantum critical points[61–63]. These kind of quantum critical points generally happens at the transition between an SSB and a non-trivial SPT phases. The key feature of these quantum critical points is that the untwisted partition functions are the same as the partition function of the transition between the SSB and a trivial symmetric phases. The differences only showed up in the twisted sectors. Roughly speaking, they can be understood as stacking an SPT state to a CFT, and one can show that there will be localized boundary states coming from the SPT state.

Here we focus on systems with $G = \mathbb{Z}_2 \times \mathbb{Z}_2$ symmetry. To study the critical point between the SSB and the SPT phases, we tune the right boundary of the sandwich to be at the transition point between $\mathcal{L}_1$ and $\mathcal{L}_5$. This theory is called Ising$^{2*}$ in Ref. [62]. We denote the corresponding state by $|\psi_{15}\rangle$. Now consider a twist defect $\mathcal{D}_s$ obtained from gauging the non-trivial SPT state, which implements the anyon permutations shown in Eq. (64). The untwisted partition function of the symmetry enriched CFT can be obtained with the help of the twist defect:

$$Z^{\mathrm{Ising}^{2*}} = \langle \mathcal{L}_1 | \psi_{15} \rangle \tag{76}$$
$$= \langle \mathcal{L}_1 | \mathcal{D}_s | \psi_{16} \rangle \tag{77}$$
$$= \langle \mathcal{L}_1 | \psi_{16} \rangle \tag{78}$$
$$= Z^{\mathrm{Ising}^2}, \tag{79}$$

where we have used the fact that $\langle \mathcal{L}_1 | \mathcal{D}_s = \langle \mathcal{L}_1 |$ in the second equality. We see that the untwisted partition function of Ising$^{2*}$ is the same as the Ising$^2$ CFT. The interpretation here is that the Ising$^2$ CFT can "absorb" the SPT state.

Using Eq. (46), one can check that the twisted partition functions $Z_{g,0}$ and $Z_{0,h}$ are also the same as the twisted partition functions in the Ising$^2$ CFT. The differences show up in the partition functions with both

spatial and temporal twists:

$$
\begin{aligned}
Z^{\text{Ising}^{2*}}_{(1,0)(0,1)} &= \langle m_1 \times \mathcal{L}_1 | W_{m_2} | \psi_{15} \rangle \\
&= \langle m_1 \times \mathcal{L}_1 | W_{m_2} \mathcal{D}_s | \psi_{16} \rangle \\
&= \langle m_1 e_2 \times \mathcal{L}_1 | W_{e_1 m_2} | \psi_{16} \rangle \\
&= -Z_{m_1} - Z_{f_1} + Z_{m_1 e_2} + Z_{f_1 e_2} \\
&= -(|\chi_1|^2 + |\chi_\psi|^2)|\chi_\sigma|^2 - (|\chi_1|^2 + |\chi_\psi|^2)(\chi_1 \overline{\chi_\psi} + \chi_\psi \overline{\chi_1}) + |\chi_\sigma|^4 + (\chi_1 \overline{\chi_\psi} + \chi_\psi \overline{\chi_1})|\chi_\sigma|^2 \\
&= -Z^{\text{Ising}^2}_{(1,0)(0,1)},
\end{aligned}
\tag{80}
$$

where, in the third equality, we push the twist defect $\mathcal{D}_s$ into the left, which implements the corresponding anyon permutation for the spatial and temporal twist, and deform the sandwich into the picture shown in Fig. 6 with the anyon $\nu_h = e_1 m_2$ wrapping around the anyon $\mu_g \times \mathcal{A} = m_1 e_2 \times \mathcal{L}_1$ threading through at the center. The minus sign in Eq. (80) is originated from the decoration of the $\mathbb{Z}_2^a$ defect line with the $\mathbb{Z}_2^b$ charge and vice versa. The twisted partition function Eq. (80) precisely measure this charge decoration.

We have seen that the untwisted partition function of the $G = \mathbb{Z}_2 \times \mathbb{Z}_2$ enriched critical point is the same as the Ising$^2$ CFT, which belongs to the same universality class as the usual Landau symmetry breaking transition. We conjecture that this is generally true for all symmetry enriched quantum critical points. The argument goes as follows. Suppose $|\Psi\rangle$ is the boundary state of a $G$ symmetry breaking transition. The partition function is given by $\langle \mathcal{A}_e | \Psi \rangle$, where $\langle \mathcal{A}_e |$ is the charge condensed boundary. The partition function of the corresponding symmetry enriched critical point between a $G$ symmetric and a $G$ SPT phases is

$$
Z^{\text{SEQCP}}_{0,0} = \langle \mathcal{A}_e | \mathcal{D}_s | \Psi \rangle,
\tag{81}
$$

where $\mathcal{D}_s$ is the SPT twist defect in the bulk of the $G$ gauge theory. It was shown in Ref. [64] that most SPT twist defects in the $G$ gauge theories implement the automorphism[65]:

$$
([g], \pi_{[g]}) \to ([g], \pi_{[g]} \times \sigma_{[g]}),
\tag{82}
$$

where $[g]$ is a conjugacy class of $G$, $\pi_{[g]}$ is an irreducible representation of the centralizer $C_g$ of $g \in G$, and $\sigma_{[g]} \in i_g \omega \in \mathcal{H}^1(G, \text{U}(1))$ is the slant product of the $(1+1)d$ SPT action $\omega \in \mathcal{H}^2(G, U(1))$, which also defines a 1d representation of the centralizer $C_g$. The automorphism Eq. (82) keeps the pure charge $([1], \pi_{[1]})$ invariant. This property implies that the charge condensed boundary is invariant under the fusion of the SPT twist defect $\mathcal{D}_s$: $\langle \mathcal{A}_e | \mathcal{D}_s = \langle \mathcal{A}_e |$. Therefore, we have

$$
Z^{\text{SEQCP}}_{0,0} = \langle \mathcal{A}_e | \mathcal{D}_s | \Psi \rangle = \langle \mathcal{A}_e | \Psi \rangle.
\tag{83}
$$

We have thus shown that the $G$ symmetry enriched quantum critical points are in the same universality class as the Landau symmetry breaking transition between the $G$ symmetric and the completely $G$ symmetry breaking phases since the universality class is determined by the untwisted partition function. However, the non-trivial SPT invariants will show up in the twisted partition functions. In general, we expect that the twisted partition function of the a CFT and its symmetry enriched version differed by the SPT torus partition function:

$$
Z^{\text{SEQCP}}_{g,h} = Z^{\text{SPT}}_{g,h} Z^{\text{CFT}}_{g,h}.
\tag{84}
$$

## C. 't Hooft anomaly and deconfined quantum critical points

Here we discuss how to use the topological holography and the sandwich picture to understand deconfined quantum critical point (DQCP) in $(1+1)$D. DQCP is a critical point with an mixed anomaly for the emergent symmetry in the IR limit. Let $G$ be the emergent IR symmetry at the critical point and $K$ be its normal subgroup that sits in the short exact sequence $1 \to K \to G \to H \to 1$ where $H = G/K$. In $(1+1)$D, the mixed anomaly is classified by $\mathcal{H}^2(K, \mathcal{H}^1(H, \text{U}(1))) \in \mathcal{H}^3(G, \text{U}(1))$. Due to this mixed anomaly, gapped phases adjacent to the DQCP generally breaks $G$ symmetry to its subgroup spontaneously.

Now we discuss an example where $G = \mathbb{Z}_2 \times \mathbb{Z}_2$, $K = \mathbb{Z}_2$, $H = \mathbb{Z}_2$. We denote the first $\mathbb{Z}_2$ symmetry by $\mathbb{Z}_2^a$ and the second one by $\mathbb{Z}_2^b$. In the sandwich construction, we choose the bulk to be a gauged SPT state

with the type-II cocycle, i.e. one in $\mathcal{H}^2(\mathbb{Z}_2, \mathcal{H}^1(\mathbb{Z}_2, \mathrm{U}(1)))$. We then have a twisted $\mathbb{Z}_2 \times \mathbb{Z}_2$ gauge theory in the bulk. We use the anyon labeling of the toric code and label the $e$ and $m$ excitations as $e_i$ and $m_i$ with $i = 1, 2$ for the first and the second copies. Unlike the usual $\mathbb{Z}_2 \times \mathbb{Z}_2$ toric code, the anyons satisfy a $\mathbb{Z}_4 \times \mathbb{Z}_4$ fusion rule, generated by $m_i$ with $m_1^2 = e_2$ and $m_2^2 = e_1$. The twisted $\mathbb{Z}_2 \times \mathbb{Z}_2$ toric code is in fact equivalent to the $\mathbb{Z}_4$ toric code after relabeling anyon. More explicitly, we can write the anyons in the twisted $\mathbb{Z}_2 \times \mathbb{Z}_2$ gauge theory in terms of the anyons in $\mathbb{Z}_4$ gauge theory as follows:

$$e_1 \to \tilde{m}^2 \tag{85}$$

$$m_1 \to \tilde{e} \tag{86}$$

$$e_2 \to \tilde{e}^2 \tag{87}$$

$$m_2 \to \tilde{m}, \tag{88}$$

where $\tilde{e}$ and $\tilde{m}$ generate the charge and the flux in the $\mathbb{Z}_4$ gauge theory respectively.

The non-trivial mutual braiding data that we are going to use are listed below:

$$\frac{S_{m_1,m_2}}{S_{0,m_j}} = i, \ \frac{S_{m_1,e_1}}{S_{0,m_1}} = -1, \ \frac{S_{m_2,e_2}}{S_{0,m_1}} = -1, \tag{89}$$

where $j = 1, 2$.

The topological symmetry of the anyon theory is $\mathbb{Z}_2 \times \mathbb{Z}_2$. The first $\mathbb{Z}_2$ is generated by

$$\sigma : m_1 \leftrightarrow m_2, e_1 \leftrightarrow e_2, \tag{90}$$

which corresponds to the EM exchange automorphism of the $\mathbb{Z}_4$ toric code. The other automorphism is given by

$$s : m_i \to m_i^3, \tag{91}$$

which comes from the automorphism of the $\mathbb{Z}_4$ group. $s$ can also be viewed as the charge conjugation symmetry of the $\mathbb{Z}_4$ toric code.

Due to the non-trivial mixed anomaly, there are only three gapped boundaries:

$$\mathcal{L}_0 = 1 \oplus m_1 \oplus e_2 \oplus m_1 e_2, \tag{92}$$

$$\mathcal{L}_1 = 1 \oplus e_1 \oplus e_2 \oplus e_1 e_2, \tag{93}$$

$$\mathcal{L}_2 = 1 \oplus m_2 \oplus e_1 \oplus m_2 e_1. \tag{94}$$

The topological defect lines on the $\mathcal{L}_0$ and $\mathcal{L}_2$ boundaries generate a $\mathbb{Z}_4$ symmetry, while on the $\mathcal{L}_1$ boundary, the symmetry is $\mathbb{Z}_2 \times \mathbb{Z}_2$.

Since we are interested in systems with $\mathbb{Z}_2 \times \mathbb{Z}_2$ symmetry, we choose the reference boundary to be $\mathcal{L}_1$. The $\mathbb{Z}_2^a$ and $\mathbb{Z}_2^b$ symmetry are generated by the $m_1$ and $m_2$ lines. Now we discuss the possible gapped phases. If we choose the $\mathcal{L}_1$ boundary on the right of the sandwich, this corresponds to a completely symmetry broken phase. The $\mathbb{Z}_2 \times \mathbb{Z}_2$ order parameters $\mathcal{O}_{\mathbb{Z}_{2a}}$ and $\mathcal{O}_{\mathbb{Z}_{2b}}$ are given by the anyon lines $e_1$ and $e_2$ across the sandwich. The degenerate ground states $|m_1\rangle$, $|m_2\rangle$, and $|m_1 m_2\rangle$ are obtained by threading $m_1$, $m_2$, and $m_1 m_2$ lines on the boundary, respectively. Due to the non-trivial mutual braiding between $m_i$ and $e_i$ as shown in Eq. (89), the expectation value of the order parameters $\mathcal{O}_{\mathbb{Z}_{2a}}$ ($\mathcal{O}_{\mathbb{Z}_{2b}}$) is opposite for $|1\rangle$ and $|m_a\rangle$ ($|m_b\rangle$).

Let's move on to the $\mathcal{L}_0$ boundary. There is only one local order parameter $\mathcal{O}_{\mathbb{Z}_{2b}}$ since only $e_2$ can condense on the $\mathcal{L}_0$ boundary but $e_1$ cannot. There are only two degenerate ground states obtained by threading $m_2$ line on the $\mathcal{L}_0$ boundary. Note that threading a $m_2^2$ line is equivalent to threading a $e_1$ line, which can be deformed and absorbed by the reference boundary. This is therefore a partially ordered phase with non-zero expectation value of $\mathcal{O}_{\mathbb{Z}_{2b}}$, which breaks $\mathbb{Z}_2^b$ symmetry.

One of the non-trivial signature of the partially ordered phase adjacent to a DQCP is that the disorder operator satisfies a $\mathbb{Z}_4$ fusion rule[66]. This can be seen rather easily from the sandwich picture. The disorder operator of the $\mathbb{Z}_2^b$ symmetry in this case corresponds to the $m_2$ anyon line across the the sandwich with a tail running on the reference boundary. The $\mathbb{Z}_4$ fusion rule of the disorder operator simply follows from the $\mathbb{Z}_4$ fusion rule of the $m_2$ anyon in the bulk.

Another non-trivial feature of the partially ordered phase is that the disorder operator of $\mathbb{Z}_2^b$ symmetry carries half of the $\mathbb{Z}_2^a$ charge[66]. This property can be understood in the sandwich picture as follows. The $\mathbb{Z}_2^a$ symmetry is generated by the $m_1$ line on the reference boundary. The $m_1$ line acts non-trivially on the disorder operator of $\mathbb{Z}_2^b$ as follows. We can move the $m_1$ line into the bulk and the symmetry action on the disorder operator is simply given by the non-trivial mutual braiding phase "$i$" between $m_1$ and $m_2$ as given in Eq. (89), which can be viewed as half of the $\mathbb{Z}_2^a$ charge.

The analysis of the $\mathcal{L}_2$ boundary is completely parallel to the discussion for the $\mathcal{L}_0$ boundary with $a \leftrightarrow b$. This choice realizes another partially ordered phase in which the local order parameter $\mathcal{O}_{\mathbb{Z}_{2a}}$ has non-zero expectation value. There is a duality between these two partially ordered phases which is generated by the automorphism $\sigma$ in Eq. (90) in the bulk.

### 1. Duality to $\mathbb{Z}_4$: the effect of changing reference boundary

In Ref. [66], the authors introduce an exact mapping between a $\mathbb{Z}_2 \times \mathbb{Z}_2$ spin chain and a $\mathbb{Z}_4$ spin chain. We now discuss this mapping in the topological holography framework.

The key point is the equivalence between the twisted $\mathbb{Z}_2 \times \mathbb{Z}_2$ Dijkgraaf-Witten theory and the $\mathbb{Z}_4$ gauge theory. Recall that the reference boundary encode the global symmetry of the systems we are interested in. In the DQCP setting, we choose $\mathcal{L}_1$ to be the reference since the global symmetry is $\mathbb{Z}_2 \times \mathbb{Z}_2$. If now we choose $\mathcal{L}_0$ to be the reference boundary, the global symmetry of the systems becomes $\mathbb{Z}_4$. This is also in some sense a "duality" between the quantum systems even though there is no automorphism defect in the bulk that connects different choices of the reference boundaries.

Let us rephrase the gapped phases in the $\mathbb{Z}_4$ toric code representation. The $\mathcal{L}_0$ becomes the electric charge condensed boundary: $\mathcal{L}_0 = 1 \oplus \tilde{e} \oplus \tilde{e}^2 \oplus \tilde{e}^3$). Choosing different gapped boundary on the right boundary of the sandwich corresponds to the following gapped phases:

$$\mathcal{L}_0 \to \mathbb{Z}_4 \text{ SSB phase} \tag{95}$$
$$\mathcal{L}_1 \to \mathbb{Z}_2 \text{ partially ordered phase} \tag{96}$$
$$\mathcal{L}_2 \to \mathbb{Z}_4 \text{ symmetric phase.} \tag{97}$$

The DQCP transition is thus related to the transition between $\mathbb{Z}_4$ SSB and $\mathbb{Z}_4$ symmetric phases. The two transitions, however, are not exactly the same. The torus partition function of the transition between $\mathbb{Z}_4$ SSB and $\mathbb{Z}_4$ symmetric phases in the anyon basis is given by

$$Z_{\text{SSB}} = \langle \mathcal{L}_0 | \Psi \rangle, \tag{98}$$
$$= Z_1 + Z_e + Z_{e^2} + Z_{e^3}. \tag{99}$$

The DQCP partition function is given by

$$Z_{\text{DQCP}} = \langle \mathcal{L}_1 | \Psi \rangle, \tag{100}$$
$$= Z_1 + Z_{e^2} + Z_{m^2} + Z_{e^2 m^2}. \tag{101}$$

Interestingly, the two partition functions are always related by a $\mathbb{Z}_2$ orbifold. To see this, we construct the twisted partition functions (twisted by the $\mathbb{Z}_2 \subset \mathbb{Z}_4$):

$$Z_{00} = Z_1 + Z_e + Z_{e^2} + Z_{e^3},$$
$$Z_{02} = Z_1 - Z_e + Z_{e^2} - Z_{e^3},$$
$$Z_{20} = Z_{m^2} + Z_{em^2} + Z_{e^2 m^2} + Z_{e^3 m^2},$$
$$Z_{22} = Z_{m^2} - Z_{em^2} + Z_{e^2 m^2} - Z_{e^3 m^2}.$$

Then it is straightforward to show that

$$Z_{\text{DQCP}} = \frac{1}{2} \left( Z_{00} + Z_{02} + Z_{20} + Z_{22} \right), \tag{102}$$

which is the $\mathbb{Z}_2$ orbifold of $Z_{\text{SSB}}$.

Let us apply our result to two examples of the transition. Ref. [66] considered the $\mathbb{Z}_4$ clock model, where the $\mathbb{Z}_4$ SSB transition is described by the $\text{Ising}^2 = \text{U}(1)_4/\mathbb{Z}_2$ CFT. Therefore, the DQCP transition is the $\text{U}(1)_4$ CFT.

We can also consider a multi-critical point for the $\mathbb{Z}_4$ SSB transition. An example is the 4-state Potts model, whose order-disorder transition is described by the $c = 1$ $\mathbb{Z}_4$ parafermion CFT. We review the $\mathbb{Z}_N$ parafermion CFT in Appendix B. We can write the partition function by using the character $\chi_{lm}$ of the $\mathbb{Z}_4$ parafermion CFT:

$$Z_{\text{SSB}} = \sum_{lm} |\chi_{lm}|^2, \tag{103}$$

where $0 \le l \le 4$, $-l + 2 \le m \le l$ with $l = m \bmod 2$ and identification $(l, m) \sim (4 - l, m \pm 4)$. Notice that the $(l, 0)$ primary for $l = 1, 2, 3$ are all $\mathbb{Z}_4$-symmetric relevant operators, so with only $\mathbb{Z}_4$ symmetry the theory describes a multicritical point. This is also consistent with the 4-state Potts model having a much larger microscopic symmetry (i.e. $S_4$) than the $\mathbb{Z}_4$ clock model.

Via the correspondence, we have a multi-critical DQCP transition Eq. (101) given by the $\mathbb{Z}_2$ orbifold of the $\mathbb{Z}_4$ parafermion CFT. It is known that the $\mathbb{Z}_4$ parafermion CFT can be represented as $\text{U}(1)_6/\mathbb{Z}_2$, so the DQCP (multicritical) transition is the $\text{U}(1)_6$ CFT. Explicit form of the DQCP partition function is given in Appendix B.

### 2. Duality between $\mathbb{Z}_2^3$ and $\mathbb{D}_8$ CFTs

In this section we study another example of DQCP. The symmetry group is $\mathbb{Z}_2^3$, with an anomaly given by the type-III cocycle $\omega$. Thus the bulk theory is the twisted gauge theory $\mathcal{Z}(\text{Vec}_{\mathbb{Z}_2^3}^\omega)$. Interestingly, the bulk theory can also be represented as a (untwisted) $\mathbb{D}_8$ gauge theory. Thus we expect that the DQCP transition is dual to the $\mathbb{D}_8$ SSB transition.

Suppose the bulk is a $G$ gauge theory $\text{D}(G)$, with anyons $([g], \pi_g)$, where $[g]$ denotes a conjugacy class, $g$ is a representative element of the conjugacy class, and $\pi_g$ is an irrep of the centralizer group $C_g$. Choose the reference boundary to be the $G$ symmetry breaking boundary, with the following Lagrangian algebra:

$$A_G = \sum_{\pi \in \text{Rep}(G)} \dim \pi \cdot ([1], \pi). \tag{104}$$

The partition function is

$$Z = \sum_{\pi \in \text{Rep}(G)} \dim \pi \cdot Z_{([1], \pi)}. \tag{105}$$

We now consider twisted partition function on a torus. Suppose there is a spatial twist labeled by the conjugacy class $[g]$. Fixing $g$, choose a conjugacy class $[h]_g$ of $C_g$ as the twist in the temporal direction, then we have a twisted partition function

$$Z^{[g],[h]_g} = \sum_{\pi_g \in \text{Rep}(C_g)} \chi_{\pi_g}(h) Z_{([g], \pi_g)}. \tag{106}$$

Let us now specialize to $G = \mathbb{D}_8 = \langle r, s | r^4 = s^2 = 1, srs = r^{-1} \rangle$. The conjugacy classes are then

$$
\begin{aligned}
[1] &= \{1\}, & C_1 &= \mathbb{D}_8 \\
[r^2] &= \{r^2\}, & C_{r^2} &= \mathbb{D}_8 \\
[r] &= \{r, r^3\}, & C_r &= \mathbb{Z}_4 = \langle r \rangle \\
[s] &= \{s, sr^2\}, & C_s &= \mathbb{Z}_2^2 = \langle s, r^2 \rangle \\
[sr] &= \{sr, sr^3\}, & C_{sr} &= \mathbb{Z}_2^2 = \langle sr, r^2 \rangle .
\end{aligned}
\tag{107}
$$

Following [64], we denote the irreps of $\mathbb{D}_8$ by $J_0, J_1, J_2, J_3$ and $\alpha$, where $J_i$'s are 1-dim, and $\alpha$ is 2-dim. We need the following fact: $\chi_{J_i}(r^2) = 1, \chi_\alpha(r^2) = -2$. The Lagrangian algebra corresponding to the $\mathbb{Z}_2^3$ duality

frame is given by

$$A_{\mathbb{Z}_2^3} = \bigoplus_i ([1], J_i) \oplus \bigoplus_i ([r^2], J_i). \tag{108}$$

Namely, it is a direct sum of all the Abelian bosons.

To get the DQCP partition function, we orbifold the $\mathbb{Z}_2$ center symmetry $\langle r^2 \rangle$. The twisted partition functions are

$$\begin{aligned}
Z^{1,1} &= Z_{(1,J_0)} + Z_{(1,J_1)} + Z_{(1,J_2)} + Z_{(1,J_3)} + 2Z_{(1,\alpha)} \\
Z^{1,r^2} &= Z_{(1,J_0)} + Z_{(1,J_1)} + Z_{(1,J_2)} + Z_{(1,J_3)} - 2Z_{(1,\alpha)} \\
Z^{r^2,1} &= Z_{(r^2,J_0)} + Z_{(r^2,J_1)} + Z_{(r^2,J_2)} + Z_{(r^2,J_3)} + 2Z_{(r^2,\alpha)} \\
Z^{r^2,r^2} &= Z_{(r^2,J_0)} + Z_{(r^2,J_1)} + Z_{(r^2,J_2)} + Z_{(r^2,J_3)} - 2Z_{(r^2,\alpha)}
\end{aligned} \tag{109}$$

Thus the orbifold partition function is

$$Z_{\text{orb}} = \frac{1}{2}(Z^{1,1} + Z^{1,r^2} + Z^{r^2,1} + Z^{r^2,r^2}) = Z_{\text{DQCP}}. \tag{110}$$

The simplest CFT with a type-III $\mathbb{Z}_2^3$ anomaly is the $\mathrm{SU}(2)_1 \simeq \mathrm{U}(1)_2$ theory. Orbifolding $\mathbb{Z}_2$ yields $\mathrm{U}(1)_8$.

### D. Intrinsically gapless SPT phases

We discuss the topological holography picture of the intrinsically gapless SPT (igSPT) phases in this section. An igSPT phase is a gapless phase that exhibiting an emergent IR anomaly in a lattice model with an anomaly-free UV symmetry[67, 68]. In general we have the following symmetry structure. Let $\Gamma$ be the microscopic on-site symmetry, which is anomaly free. Below an energy scale $\Delta$, there is an emergent IR symmetry $G$ that sits in the short exact sequence:

$$1 \to H \to \Gamma \to G \to 1, \tag{111}$$

where $H$ is a normal subgroup of $\Gamma$ which acts trivially on the IR degrees of freedom, and $G = \Gamma/H$. The emergent IR symmetry $G$ has an anomaly in $\mathcal{H}^{D+1}(G, \mathrm{U}(1))$, where $D$ is the spacetime dimensions. This anomaly becomes trivial when it is pulled back into $\mathcal{H}^{D+1}(\Gamma, \mathrm{U}(1))$, so the entire system can be realized in $D$ dimensions with non-anomalous $G$ symmetry.

To illustrate the main picture, we focus on the $(1+1)d$ igSPT phases with $\Gamma = \mathbb{Z}_4$, $H = \mathbb{Z}_2$, and $G = \mathbb{Z}_2$. The non-trivial igSPT phase realized in the $\Gamma = \mathbb{Z}_4$ lattice model shares the same (emergent) anomaly as the gapless edge modes of the $\mathbb{Z}_2$ Levin-Gu SPT state. The holographic picture of this igSPT state is shown in Fig. 7. The bulk of the sandwich consists of a $\mathbb{Z}_4$ toric code topological order on the left and a double semion order on the right. The two topological orders are separated by a gapped domain wall on which the $e^2m^2$ particle in the $\mathbb{Z}_4$ toric code condenses. We will denote the domain wall by $\mathcal{I}_{e^2m^2}$. After the condensation, the deconfined anyons in the $\mathbb{Z}_4$ toric code are $\{[1], [em], [em^3], [m^2]\}$, which can be identified with the anyons in the double semion order $\{1, s, \bar{s}, b\}$, respectively.

The left topological gapped boundary is the $e$ condensed boundary of the $\mathbb{Z}_4$ toric code. The confined anyons on the left boundary are generated by the $m$ particle, which satisfies a $\mathbb{Z}_4$ fusion rule. The $m$ line on the left boundary thus generates the $\mathbb{Z}_4$ global symmetry in the sandwich picture. On the right non-topological boundary of the double semion order, we have the edge theory of the $\mathbb{Z}_2$ Levin-Gu SPT state. Upon dimensional reduction, the sandwich construction realizes the igSPT state where the low-energy theory is described by the gapless edge of the $\mathbb{Z}_2$ Levin-Gu SPT state in a system with a $\mathbb{Z}_4$ global symmetry.

We can use the sandwich picture to understand the following three main properties of the $\mathbb{Z}_4$ igSPT state:

1. There exists a low-energy subspace such that the end of the $\mathbb{Z}_4$ symmetry defect line is charged under the $\mathbb{Z}_2$ normal subgroup.

2. There exists a low-energy subspace such that the end of the $\mathbb{Z}_2$ symmetry defect line is charged under the $\mathbb{Z}_4$ symmetry.

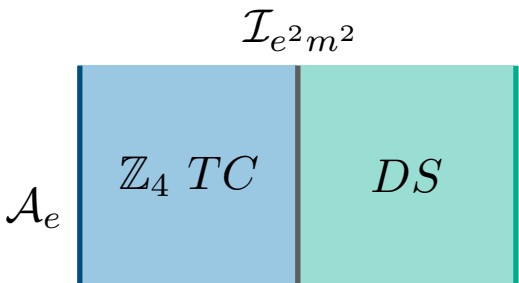

FIG. 7. The sandwich picture of the $\mathbb{Z}_4$ igSPT state. The bulk of the sandwich is formed by a $\mathbb{Z}_4$ toric code ($\mathbb{Z}_4$ TC) on the left and a double semion (DS) order on the right, separated by a gapped domain wall $\mathcal{I}_{e^2m^2}$ on which the $e^2m^2$ particle in the $\mathbb{Z}_4$ toric code condenses. The left topological boundary is the $e$ condensed boundary. The right boundary supports the edge modes of the $\mathbb{Z}_2$ Levin-Gu SPT state.

Now we discuss the first property. Fig. 8 (a) shows an open $\mathbb{Z}_4$ symmetry defect line with a defect operator $\mathcal{O}_1$ attached to the end. The $\mathbb{Z}_4$ symmetry $U_1$ is implemented by the $m$ line confined on the left topological boundary. The defect operator $\mathcal{O}_1$ is attached to the end of the $\mathbb{Z}_4$ defect line $m$, which corresponds to the bulk anyon line $em$ in the $\mathbb{Z}_4$ toric code, attaching to a semion line $s$ in the double semion order at the gapped domain wall $\mathcal{I}_{e^2m^2}$. Note that the defect operator should correspond to the anyons that can freely pass through the gapped domain wall $\mathcal{I}_{e^2m^2}$ since, after dimensional reduction, they become operators localized at the end of the symmetry defect line. In the IR limit, the $m$ line becomes a $s$ line which generates the $G = \mathbb{Z}_2$ symmetry.

The $H = \mathbb{Z}_2$ symmetry in the $\mathbb{Z}_2$ normal subgroup is implemented by a $m^2$ line on the left boundary, which can be moved into the $\mathbb{Z}_4$ toric code bulk and becomes an $e^2m^2$ line. Since $e^2m^2$ condenses on the domain wall $\mathcal{I}_{e^2m^2}$, it only acts on the gapped degrees of freedom, and acts as the identity in the IR limit (in the double semion side). However, there exist a low-energy subspace such that an $m^2$ line becomes an $e^2m^2 \times e^2 = m^2$ line in the bulk of $\mathbb{Z}_4$ toric code, and then becomes the $b$ line in the double semion bulk[69]. Within this low-energy subspace, the $\mathbb{Z}_2$ symmetry generated by the bulk $m^2$ line then acts non-trivially on the defect operator of the $\mathbb{Z}_4$ symmetry, due to the non-trivial mutual braiding between $m^2$ and $em$ particles as shown in Fig. 8 (a). Therefore, we see that the $\mathbb{Z}_4$ defect operator is charged under the $\mathbb{Z}_2$ symmetry in the low-energy subspace.

For the second property, we consider the defect operator $\mathcal{O}_2$ of the $\mathbb{Z}_2$ symmetry $U_2$ in the low-energy subspace defined above. It is represented in the $\mathbb{Z}_4$ toric code as an $m^2$ line meeting with a $b$ line in the double semion order at the gapped domain wall $\mathcal{I}_{e^2m^2}$, as shown in Fig. 8 (b). The generator of the $\mathbb{Z}_4$ symmetry is given by the $m$ line on the left boundary. When we move the $m$ line into the bulk of the $\mathbb{Z}_4$ toric code, it becomes an $em$ line, and then becomes a semion line $s$ in the double semion order. Due to the non-trivial mutual braiding between $em$ and $m^2$ (or $s$ and $b$) particles, the $\mathbb{Z}_2$ defect operator is charged under the $\mathbb{Z}_4$ symmetry in the low-energy subspace.

Now we discuss the partition function of the igSPT phases. For concreteness, we take the right boundary to be the $U(1)_2$ CFT [70]. The partition functions in the DS sectors are given by:

$$
\begin{aligned}
Z_1 &= |\chi_0|^2, \\
Z_s &= \chi_1 \overline{\chi}_0, \\
Z_{\bar{s}} &= \chi_0 \overline{\chi}_1, \\
Z_b &= |\chi_1|^2.
\end{aligned}
\tag{112}
$$

The gapped domain wall $\mathcal{I}_{e^2m^2}$ in the sandwich picture can be written as

$$
\mathcal{I}_{e^2m^2} = \sum_{\alpha \in \mathbb{Z}_4 \mathrm{TC}, \mu \in \mathrm{DS}} W_{\alpha,\mu} |\alpha\rangle \langle \mu|.
\tag{113}
$$

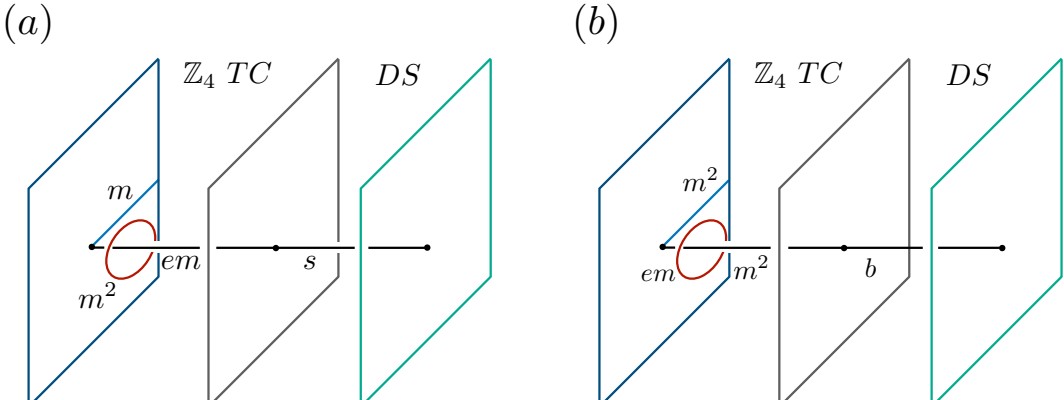

FIG. 8. The defect operators for $\mathbb{Z}_4$ igSPT phases. (a) The $\mathbb{Z}_4$ defect operator $\mathcal{O}_1$ is given by the anyon line $em$ in the bulk of the a $\mathbb{Z}_4$ toric code connecting to a $s$ line in the double semion theory. Due to the non-trivial braiding between the $m^2$ (red loop) and $em$ (or $b$ and $s$ if we move $m^2$ loop into the double semion bulk), the defect operator $\mathcal{O}_1$ is charged under the $\mathbb{Z}_2$ normal subgroup. (b) The $\mathbb{Z}_4$ defect operator $\mathcal{O}_2$ is given by the anyon line $m^2$ in the bulk of the a $\mathbb{Z}_4$ toric code connecting to a $b$ line in the double semion theory. The defect operator $\mathcal{O}_2$ is charged under the $\mathbb{Z}_4$ symmetry due to the non-trival braiding between $em$ (red loop) and $m^2$ lines.

The untwisted torus partition function can then be obtained from the sandwich:

$$
\begin{aligned}
Z_{00} &= \langle \mathcal{A}_e | \mathcal{I}_{e^2 m^2} | \Psi \rangle \\
&= \sum_{\alpha \in \mathbb{Z}_4 \mathrm{TC}, \mu \in \mathrm{DS}} w_\alpha W_{\alpha,\mu} Z_\mu \\
&= Z_1 + Z_b,
\end{aligned}
\tag{114}
$$

which is the same as the usual $\mathrm{U}(1)_2$ CFT. The non-trivial characterization is in the twisted partition functions since we can twist by the element in $H$. Specifically, consider the partition functions with a $g \in G = \mathbb{Z}_2$ twist in the space direction and a $h \in H = \mathbb{Z}_2$ twist in the time direction:

$$
Z_{g,h} = \mathrm{Tr}_{\mathcal{H}_g} h e^{-\beta \mathcal{H}}.
\tag{115}
$$

Recall that the $g$ twist corresponds to an $m$ line on the left boundary that generates the $\Gamma = \mathbb{Z}_4$ symmetry. The $h$ twist corresponds to the $m^2$ line (which can be moved into the $\mathbb{Z}_4$ TC bulk and remains $m^2$ line) that generates the $\mathbb{Z}_2$ normal subgroup in $\Gamma = \mathbb{Z}_4$. The twisted partition function can be calculated in the sandwich picture as

$$
\begin{aligned}
Z_{g,h} &= \sum_{\alpha \in m \times \mathcal{L}_e, \mu \in \mathrm{DS}} \frac{S_{\alpha,m^2}}{S_{0,\alpha}} W_{\alpha,\mu} Z_\mu \\
&= -(Z_s + Z_{\bar{s}}) \\
&= -Z_{g,0},
\end{aligned}
\tag{116}
$$

where $m \times \mathcal{L}_e = (m, em, e^2 m, e^3 m)$. The minus sign in Eq. (116) is the $h$ charge decorated on the $g$ defect line, and the twisted partition function $Z_{g,h}$ precisely measure this charge decoration. Following the argument in Ref. [67], one can show that the $h$ defect line is also charged under the $g$ symmetry in the "low temperature" limit, i.e. $\beta \to \infty$, by using the $S$ transformation:

$$
Z_{h,g}(\beta) = Z_{g,h}(1/\beta) = -Z_{g,0}(1/\beta).
\tag{117}
$$

In the limit $\beta \to \infty$, the minus sign is the leading term of the twisted partition function $Z_{h,g}(\beta)$.

## V.  BOUNDARY STATES IN RCFTS

We will now apply the topological holography framework to boundary conformal field theories, to provide holographic interpretations of Cardy and Ishibashi boundary states for RCFTs.

### A.  Review of BCFTs

In this section, we review basic facts about boundary states in RCFTs [71, 72]. There are two pictures for boundary CFTs. In the open-channel picture, one considers the partition function of a CFT on an interval of length $L$, with conformal boundary conditions $A$ and $B$ on the two ends. Define the Euclidean partition function

$$\mathcal{Z}_{AB} = \mathrm{Tr}_{\mathcal{H}_{AB}} e^{-\beta H}. \tag{118}$$

Here $\mathcal{H}_{AB}$ is the CFT Hilbert space with the given boundary conditions.

To get the closed-channel picture, we perform a space-time rotation on the cylinder, and view $\mathcal{Z}_{\mathcal{AB}}$ as evolving the system defined on the "time" circle of circumference $\beta$ from boundary states $A$ to $B$:

$$\mathcal{Z}_{AB} = \langle A| e^{-LH} |B \rangle. \tag{119}$$

It turns out to be most convenient to describe boundary conditions using boundary states.

In general, a conformal boundary condition only preserves one copy of the Virasoro symmetry. For simplicity, let us assume that the boundary conditions preserve one copy of the chiral algebra. Then the Hilbert space $\mathcal{H}_{AB}$ admits the following decomposition:

$$\mathcal{H}_{AB} = \bigoplus_a n_{AB}^a \mathcal{H}_a, \tag{120}$$

and the partition function

$$\mathcal{Z}_{AB} = \sum_a n_{AB}^a \chi_a(q). \tag{121}$$

The non-negative integers $n_{ab}^a$ give the operator content with the boundary conditions $A, B$.

A complete basis for boundary states is given by the Ishibasha states:

$$|h\rangle\rangle = \sum_{n=0}^{\infty} \sum_{j=1}^{d_h(n)} |n, j; h\rangle_L \otimes |n, j; h\rangle_R. \tag{122}$$

Here $h$ labels a chiral primary operator, and $|n, j; h\rangle_{L/R}$ are the descendant states in the conformal tower labeled by $h$ with dimension $h + n$, with $j$ labeling the $d_h(n)$-fold degeneracy of the corresponding state. However, the Ishibashi states are not physical boundary states in the sense that the partition function with Ishibasha boundary states does not take the desired form in (121).

Physical boundary states can be constructed from linear superpositions of Ishibashi states. For diagonal RCFTs, a complete set of physical boundary states that preserves one copy of the chiral algebra are constructed by Cardy: for each primary $a$ there is a corresponding Cardy state $|a\rangle$, given by

$$|a\rangle = \sum_b \frac{S_{ab}}{\sqrt{S_{0b}}} |b\rangle\rangle. \tag{123}$$

From Verlinde formula, we find the multiplicity $n_{ab}^c = N_{ab}^c$. In particular, $n_{ab}^1 = \delta_{a\bar{b}}$.

For minimal models (with diagonal partition functions), the Cardy states give all physical boundary states. For RCFTs with extended chiral algebra, there can be physical boundary states that break the chiral algebra. Fully classifying boundary states for RCFTs is a challenging problem and remains open to date.

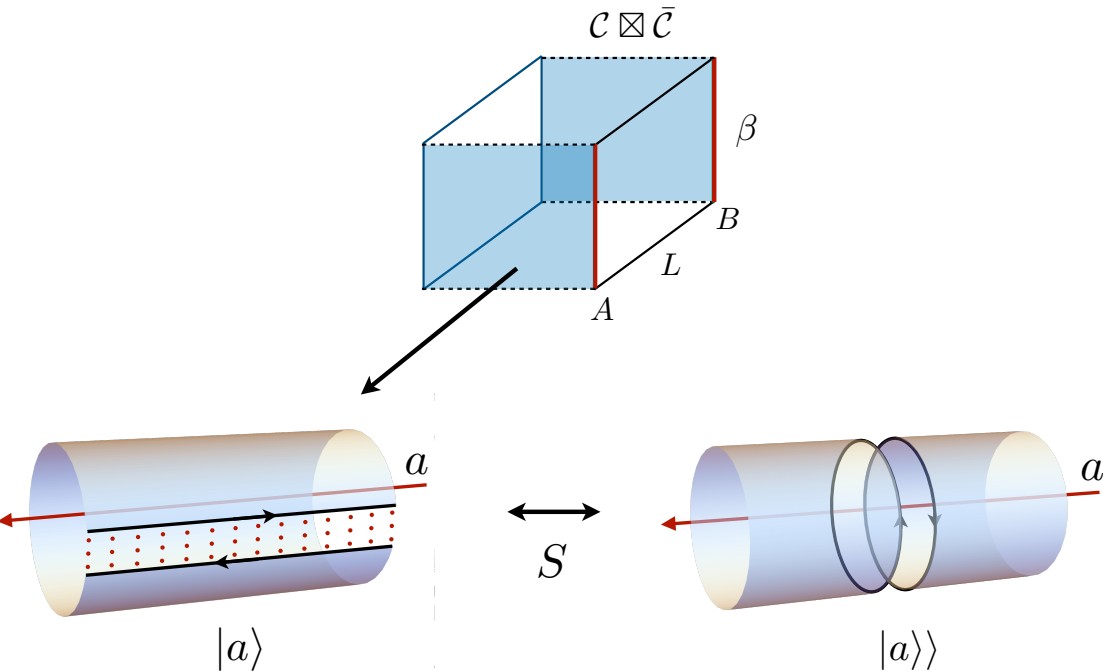

FIG. 9. Holographic picture for the boundary states. The top figure shows the sandwich construction of a diagonal RCFT with boundary states denoted as $A$ and $B$ (red lines). The bulk of the sandwich is $\mathcal{C} \boxtimes \overline{\mathcal{C}}$. In the space-time rotated picture, the direction of time evolution is going from the boundary state $A$ to $B$ defined on a circle of circumference $\beta$. Through unfolding, the blue planes near $t = 0$ and $t = L$ can be viewed as a quantum quench problem of a $(2+1)$D chiral topological order with MTC $\mathcal{C}$. The Cardy states then correspond to coupling the edge chiral CFTs to form a torus with different anyon flux threading through as shown in the lower left figure. After the $S$ transformation, we obtain a different quantum quench problem where the cut is perpendicular to the anyon threading through as shown in the lower right figure. Coupling the edge CFTs in this case correspond to the Ishibashi states.

## B. Holographic perspective

In this section, we provide a simple physical interpretation of the boundary states based on the holographic description. For simplicity, let us first consider a diagonal RCFT, whose associated MTC is $\mathcal{C}$. As reviewed in Sec. V A, we can view the partition function $\mathcal{Z}_{AB}$ as the time evolution from boundary state $A$ at $t = 0$ to $B$ at $t = L$ with circumference $\beta$. In the topological holographic picture, we choose the bulk to be $\mathcal{Z}(\mathcal{C}) = \mathcal{C} \boxtimes \overline{\mathcal{C}}$. The reference topological boundary is the canonical one $\oplus_{a \in \mathcal{C}}(a, a)$.

At time immediately before $t = 0$ (or immediately after $t = L$), the system is in a gapped phase. This gapped phase corresponds to a choice of the gapped boundary condition for the right boundary in the sandwich construction. The simplest and most obvious choice is again the canonical boundary $\oplus_{a \in \mathcal{C}}(a, a)$. This leads to a gapped phase with degenerate ground states with dimension $|\mathcal{C}|$, and can be labeled by the anyons in $\mathcal{C}$. However, this gapped phase is not always accessible by some relevant operators. If we consider the gapped phase that is driven by a relevant operator, this relevant operator could cause the splitting of the ground state degeneracy. In this case, it proves to be more instructive to think of the system as a strip (annulus) of a $(2+1)d$ chiral phase described by the MTC $\mathcal{C}$. For diagonal RCFTs, the canonical boundary becomes the completely trivial domain wall in the middle of the strip (annulus). Choosing a gapped boundary means that the gapless edges of the strip (annulus) described by the chiral CFT are glued together such that the strip becomes a cylinder (torus) as shown in Fig. 9. Therefore, the boundary states can be viewed as coupling the edge chiral CFTs to form a cylinder (torus) with different anyon flux threaded, which can naturally be understood as gapped phases of $(1+1)d$ systems.

We now discuss the gluing process and the boundary states in more details. The discussion below will be focused on the boundary states, and the holographic derivation of the boundary partition function is in Appendix. V C. Let us focus on the boundary state $A$, the gluing process can be viewed as a quantum quench

problem with the following Hamiltonian:

$$H = H_L + H_R + gH_{\text{int}}, \tag{124}$$

where $H_L$ and $H_R$ denote the Hamiltonians of left/right-moving edge states, respectively, and $H_{\text{int}}$ a RG relevant interedge coupling. We start at $t = 0$ from the Hamiltonian with the interedge coupling turned on, and then switched off completely. The initial fully gapped state corresponds to the conformally invariant boundary state $A$. The boundary state is then mapped to a state of the topological phase on a torus with possible insertion of anyons $a$. We denotes the corresponding boundary state as $|a\rangle$. We will see that this boundary state is the Cardy state.

The other set of closely related boundary states, the Ishibashi states, can be obtained by performing a $S$ transformation. After the $S$ transformation, the quantum quench problem is defined on the same torus but with a different cut as shown in the lower right figure in Fig. 9. In this case, the cut is glued back such that the anyon line threading through the torus is perpendicular to the cut. This problem has been studied in Ref. [73–76], and it has been shown that the initial gapped state corresponds to the Ishibashi state:

$$|a\rangle\rangle = \sum_{n=0}^{\infty} \sum_{j=1}^{d_a(n)} |k(a,n), j; a\rangle_L \otimes |-k(a,n), j; a\rangle_R. \tag{125}$$

Here $k(a,n) = 2\pi(h_a + n)/L_y$ denotes the momentum, where $L_y$ is the circumference of the boundary circle; $j$ labels the elements of an orthonormal basis in the subspace of fixed momentum $k(a,n)$.

We note that there is an issue related to the Ishibashi states $|a\rangle\rangle$ being non-normalizable. To make sense it is reasonable to introduce some high energy cut-off. In the case of the topological phase, the cut off is given by the bulk energy gap. Here a more convenience choice is to introduce a "temperature" and consider the smeared states:

$$e^{-\frac{\tau}{2}(H_L+H_R)}|a\rangle\rangle. \tag{126}$$

The overlap is given by

$$\langle\langle a|e^{-\frac{\tau}{2}(H_L+H_R)}|a\rangle\rangle = \chi_a(e^{-4\pi\tau}). \tag{127}$$

We can use $S$ transformation to write

$$\chi_a(e^{-4\pi\tau}) = \sum_b S_{ab}\chi_b(e^{-\pi/\tau}). \tag{128}$$

We are interested in the limit $\tau \to 0$ (i.e. high-temperature) so that

$$\chi_b(e^{-\pi/\tau}) \cong e^{-\frac{\pi}{\tau}(h_b - \frac{c}{24})}. \tag{129}$$

The vacuum sector $b = 0$ dominates the sum, therefore we obtain

$$\langle\langle a|e^{-\frac{\tau}{2}(H_L+H_R)}|a\rangle\rangle \cong S_{a0}e^{\frac{\pi c}{24\tau}}. \tag{130}$$

While the $\tau \to 0$ limit is formally divergent, which is a consequence of the high-temperature density of states (Cardy's formula), the relative normalization between different $a$'s are given by $\sqrt{S_{a0}}$. In other words, the normalized states

$$\frac{|b\rangle\rangle}{\sqrt{S_{0b}}} \tag{131}$$

now have a uniform norm independent of $b$. We will therefore use Eq. (131) as the properly normalized states.

As discussed above, the original initial state $|a\rangle$ of the quantum quench is related to Eq. (131) by a $S$ transformation:

$$|a\rangle = \sum_b \frac{S_{ab}}{\sqrt{S_{0b}}}|b\rangle\rangle, \tag{132}$$

which is precisely the relations between the Cardy and Ishibashi states. Therefore, boundary states can be viewed as coupling the edge CFTs on the cut to form a torus with different anyon flux threaded.

This holographic picture naturally explains why Cardy states are physical. As a gapped state in one dimension, the Cardy state with an anyon threaded in the long direction of the cylinder has only short-range correlations (i.e. satisfy cluster decomposition). Recall that local operators correspond to Wilson lines connecting the two edges, which become Wilson loops along the compactified direction after gluing. The Cardy states are eigenstates of the Wilson loops. In contrast, Ishibashi states, being superpositions of Cardy states, are generally long-range correlated (i.e. do not obey cluster decomposition).

This physical picture can be generalized to non-diagonal RCFTs. For non-diagonal RCFTs, there is a non-trivial domain wall in the middle of the strip, which can be viewed as having a thin strip of a topological order described by another MTC $\mathcal{D}$. The two MTCs $\mathcal{C}$ and $\mathcal{D}$ are separated by a gapped domail wall described the module category $\mathcal{T}$ [77]. Physically, $\mathcal{D}$ is obtained by condensing an algebra $\mathcal{A}$ in $\mathcal{C}$. The particles living on the gapped domain wall are described by the module category $\mathcal{T}$, which contains confined and deconfined anyons in the condensed phase $\mathcal{D}$. Similar to the case of diagonal RCFTs, the boundary states can then be viewed as coupling the CFTs to form a torus in the presence of this defect. The Cardy states then correspond to inserting anyons that can freely tunnel through the defect. Hence, they are labeled by the deconfined anyons in $\mathcal{D}$. The most general object that can be inserted to the torus is particles in the gapped domain wall $\mathcal{T}$. Some of them are mapped to the confined anyons, which correspond to the non-Cardy boundary states. Therefore, elementary boundary states for general RCFTs are labeled by the simple objects in the module category $\mathcal{T}$.

## C. BCFT partition functions from holography

We now explain the holographic derivation of the BCFT partition function for RCFTs. For diagonal RCFTs, it is a well-known result that

$$\mathcal{Z}_{ab} = \sum_{c \in \mathcal{C}} N_{ab}^c \chi_c(q), \tag{133}$$

where $q = \exp(-\pi\beta/L)$, and $a, b, c \in \mathcal{C}$. Consider a diagonal RCFT defined on a cylinder with length $L$ and circumference $\beta$ with boundary conditions $a$ and $b$, which are Cardy states. We go through the sandwich construction with the bulk being $\mathcal{C} \boxtimes \bar{\mathcal{C}}$ such that the RCFT lives on the right boundary and we choose the left topological boundary condition to be the $\mathcal{A} = \oplus_i(i, i)$ condensed boundary. Fig. 10 shows the picture for general RCFTs. For diagonal RCFTs, we simply choose the gapped domain wall $\mathcal{T}$ to be the trivial domain wall. To proceed, we shrink the left topological boundary so that we obtain an solid cylinder consisted of the bulk topological order $\mathcal{C} \boxtimes \bar{\mathcal{C}}$ with a Lagrangian algebra anyon $\mathcal{A} = \oplus_i(i, i)$ threading through the bulk. The solid cylinder can be unfolded into a solid torus with major circumference being $2L$. The bulk topological order is $\mathcal{C}$ with a direct sum of anyons $\oplus i \in \mathcal{C}$ threading through. The boundary of the solid torus is the chiral CFT with characters $\chi_i(\tilde{q})$, where $\tilde{q} = \exp(-4\pi L/\beta)$. For diagonal RCFTs, $a$ and $b$ are simple objects in the category $\mathcal{C}$. Using the fusion rule to fuse the loops $a$ and $b$: $a \times b = \oplus_c N_{ab}^c c$, and the definition of the $S$ matrix, the boundary CFT partition function is then given by

$$\begin{aligned}
\mathcal{Z}_{ab} &= \sum_c \sum_i (\sqrt{S_{0i}})^2 N_{ab}^c \frac{S_{ci}}{S_{0i}} \chi_i(\tilde{q}) \\
&= \sum_c \sum_i N_{ab}^c S_{ci} \chi_i(\tilde{q}),
\end{aligned} \tag{134}$$

where the $(\sqrt{S_{0i}})^2$ factor is the normalization of the TQFT partition function involving the boundary states[78]. Using the modular transformation of the characters:

$$\chi_c(q) = \sum_i S_{ci} \chi_i(\tilde{q}), \tag{135}$$

we obtain Eq. (133).

We now generalize the derivation to non-diagonal RCFTs. Recall that our holographic picture for a non-diagonal RCFT is a strip of the chiral topological phase $\mathcal{C}$, with a thin strip of $\mathcal{D}$ topological order inserted in

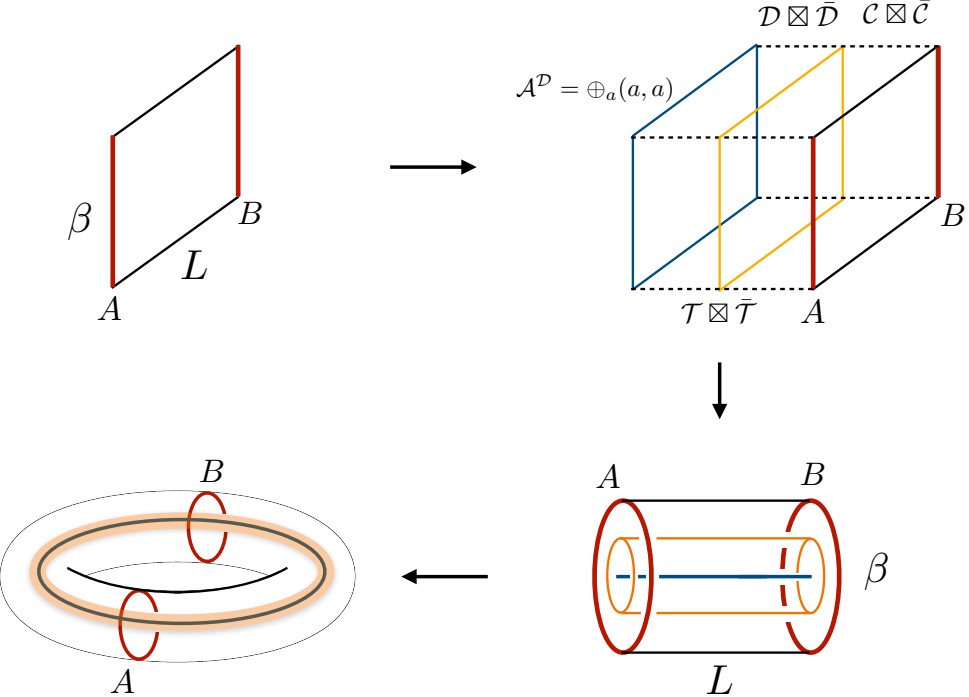

FIG. 10. Derivation of the boundary partition function for RCFTs based on hologrphic picture.

the middle as a topological defect. Boundary states correspond to objects in the module category $\mathcal{T}$, which describes the gapped domain wall between $\mathcal{C}$ and $\mathcal{D}$. This can be "folded" into a sandwich construction, where the left topological gapped boundary can be viewed as a composition of a gapped domain wall between $\mathcal{C} \boxtimes \bar{\mathcal{C}}$ and $\mathcal{D} \boxtimes \bar{\mathcal{D}}$ and a canonical $\mathcal{A}^{\mathcal{D}} = \oplus_a (a, a)$ condensed boundary in $\mathcal{D} \boxtimes \bar{\mathcal{D}}$. We then shrink the left topological boundary to obtain a solid cylinder with a coaxial gapped domain wall separating $\mathcal{C} \boxtimes \bar{\mathcal{C}}$ and $\mathcal{D} \boxtimes \bar{\mathcal{D}}$, and with a Lagrangian algebra anyon $\oplus_a (a, a)$ threading through. We unfold this solid cylinder and obtain a solid torus with a gapped domain wall $\mathcal{T}$ separating $\mathcal{C}$ and $\mathcal{D}$ with an insertion of anyon $\oplus_a a \in \mathcal{D}$ as shown in Fig. 10. The particles in the gapped domain wall are described by a theory $\mathcal{T}$, which is a category $\mathcal{C}_{\mathcal{A}}$ of $\mathcal{A}$-module in $\mathcal{C}$, and $\mathcal{A}$ is a $\mathcal{C}$ algebra corresponding to the condensed anyon in $\mathcal{C}$. In general, boundary states correspond to simple objects in $\mathcal{T}$.

Suppose the boundary states are given by $A = s \in \mathcal{T}$ and $B = t \in \mathcal{T}$. We first apply the fusion rule in the theory $\mathcal{T}$ to fuse the boundary states: $s \times t = \oplus_u (N^{\mathcal{T}})_{st}^u u$. We lift $u$ into an object in $\mathcal{C}$: $\oplus_i (B^{\mathcal{T}|\mathcal{C}})_{u,i} i \in \mathcal{C}$, where $(B^{\mathcal{T}|\mathcal{C}})_{u,i}$ is the lifting coefficient[79] (also called a branching matrix[55, 56]). We also lift each anyon $a$ inserted in the middle of the solid torus into an object in $\mathcal{C}$: $\oplus_j (B^{\mathcal{D}|\mathcal{C}})_{a,j} j \in \mathcal{C}$. Using the definition of the $S$ matrix in $\mathcal{C}$, we obtain the boundary partition function:

$$
\begin{aligned}
\mathcal{Z}_{st} &= \sum_{u,a,i,j} (n^{\mathcal{T}})_{st}^u (B^{\mathcal{T}|\mathcal{C}})_{u,i} (B^{\mathcal{D}|\mathcal{C}})_{a,j} S_{ij}^{\mathcal{C}} \chi_j^{\mathcal{C}}(\tilde{q}) \\
&= \sum_{u,a,i,j} (n^{\mathcal{T}})_{st}^u (B^{\mathcal{T}|\mathcal{C}})_{u,i} (B^{\mathcal{D}|\mathcal{C}})_{a,j} S_{ij}^{\mathcal{C}} (S^{\mathcal{C}})_{jk}^{-1} \chi_k^{\mathcal{C}}(q).
\end{aligned}
\tag{136}
$$

Here we apply the modular transformation of the characters in $\mathcal{C}$:

$$
\chi_b^{\mathcal{C}}(q) = \sum_a S_{ab}^{\mathcal{D}} \chi_a^{\mathcal{C}}(\tilde{q}).
\tag{137}
$$

Generalizing the diagonal case, we conjecture that Eq. (136) can be written as follows:

$$
\mathcal{Z}_{st} = \sum_{u,i} (n^{\mathcal{T}})_{st}^u (B^{\mathcal{T}|\mathcal{C}})_{u,i} \chi_i^{\mathcal{C}}(q).
\tag{138}
$$

We have verified the conjecture for many examples.

The boundary partition functions for the Cardy states can be obtained by restricting $s$ and $t$ in Eq. (138) to be the particles that map to deconfined anyons in $\mathcal{D}$.

## D. Boundary states and renormalization group flows

To understand the nature of the gapped phases corresponding to the boundary states, we consider the gapped phase is obtained from a CFT perturbed by a relevant operator $\Phi$:

$$H = H_{\text{CFT}} + \lambda \int \Phi(x) d^2 x. \tag{139}$$

We assume that this relevant operator takes the CFT to a gapped phase, which corresponds to a Cardy state $|A\rangle$. As discussed in the previous section, using the bulk picture, we can think of the boundary state as the dimensional reduction of a $(2+1)$D chiral topological order on a long cylinder with anyon flux threading through. The Wilson loop operator $W_b$ along the compactified direction becomes a local operator in the $(1+1)$D system. The expectation value of the Wilson loop operator is given by the mutual braiding:

$$\langle a|W_b|a\rangle = \frac{S_{ab}}{S_{0a}}. \tag{140}$$

In the gapped phase perturbed by a relevant operator $\Phi_b$, we should have $\langle \Phi_b \rangle \neq 0$. Eq. (140) provides a easy way to calculate this expectation value by the identification $\langle \Phi_b \rangle = \langle W_b \rangle$. We can also compute the expectation value of the order parameter by using Eq. (140), which is helpful to identify the symmetry breaking phases.

In practice, we may find there are several possible boundary states for a given sign of the coupling. To determine which one survives in the low energy limit, we can use Cardy's variational approach[49] to see which boundary states have the lowest energy. The general expression of the variational energy is given by

$$E_a = \frac{\pi c}{24(2\tau_a)^2} + \frac{S_{ab}}{S_{0a}} \frac{\tilde{\lambda}_b}{(2\tau_a)^{\Delta_b}}, \tag{141}$$

where $\tau_a$ is the cut-off in the variational ansatz, $\tilde{\lambda}_b$ is the rescaled coupling $\tilde{\lambda}_b = \pi^{\Delta_b}(S_{00}/S_{b0})^{1/2}\lambda_b$, $\Delta_b$ is the scaling dimension of the coupling $\Phi_b$. We see that the same factor $S_{ab}/S_{0a}$ appears in Eq. (141). Therefore, the expectation value of the Wilson loop $\langle a|W_b|a\rangle$ can also help us to find the minimum variational energy among the possible boundary states.

## E. Examples

### 1. Ising model

We begin the discussion with the simplest example: Ising CFT. The bulk topological order is given by the Ising category with anyons $\{1, \sigma, \psi\}$. It is known that all boundary states are Cardy states.

We consider the $\psi$ perturbation. Vising from the bulk picture, this is a $W_\phi$ Wilson loop winding along the compactified direction. In the $\langle W_\psi \rangle > 0$ phase, we find two degenerate Cardy states $|1\rangle$ and $|\psi\rangle$ have the lowest energy by using Eq. (141). To see the nature of the gapped phase, we note that the local $\mathbb{Z}_2$ order parameter corresponds to the $W_\sigma$ Wilson loop winding along the compactified direction. Using Eq. (140), we see the $\mathbb{Z}_2$ order parameter $\langle W_\sigma \rangle$ has opposite eigenvalues in these two states, indicating that we are in the $\mathbb{Z}_2$ spontaneously broken phase.

In the $\langle W_\psi \rangle < 0$ phase, on the other hand, there is a unique ground state given by $|\sigma\rangle$. One can check that $\langle W_\sigma \rangle = 0$ so that we are in the $\mathbb{Z}_2$ symmetric phase.

## 2. Tricritical Ising model

Now we apply this approach to the tricritical Ising CFT. The tricritical Ising CFT has six primaries:

$$1_{0,0}, \ \epsilon_{\frac{1}{10},\frac{1}{10}}, \ \epsilon'_{\frac{3}{5},\frac{3}{5}}, \epsilon''_{\frac{3}{2},\frac{3}{2}}, \sigma_{\frac{3}{80},\frac{3}{80}}, \sigma'_{\frac{7}{16},\frac{7}{16}}. \tag{142}$$

In the Landau-Ginzburg picture, they correspond to the scalar fields:

$$1, \ \phi^2, \ \phi^4, \ \phi^6, \ \phi, \ \phi^3, \tag{143}$$

respectively. There is a $\mathbb{Z}_2$ invertible line $\eta$ which only acts non-trivially on $\sigma$ and $\sigma'$, and four non-invertible Verlinde lines generated by $\mathcal{W}$ and $\mathcal{N}$.

The bulk topological order corresponding to the tricritical Ising CFT is described by the stacking of a $\mathrm{Spin}(7)_1$ theory and a time-reversal conjugation of a Fibonacci theory: $\mathrm{Spin}(7)_1 \times \overline{\mathrm{Fib}}$ [80]. We label the bulk anyon by a tuple $a = (a^I, a^F)$, where $a^I \in \{1, \sigma, \psi\}$ and $a^F \in \{1, \epsilon\}$. The correspondence of the primaries and the bulk anyons is given below:

$$1 \leftrightarrow (1,1), \ \epsilon' \leftrightarrow (1,\epsilon), \ \sigma' \leftrightarrow (\sigma,1) \tag{144}$$

$$\sigma \leftrightarrow (\sigma,\epsilon), \ \epsilon'' \leftrightarrow (\psi,1), \ \epsilon \leftrightarrow (\psi,\epsilon). \tag{145}$$

The S-matrix is given by $S_{ab} = S_{a^I,b^I} S_{a^F,b^F} =$

$$\frac{1}{2\sqrt{\xi^2+1}} \begin{pmatrix} 1 & \xi & \sqrt{2} & \sqrt{2}\xi & 1 & \xi \\ \xi & -1 & \sqrt{2}\xi & -\sqrt{2} & \xi & -1 \\ \sqrt{2} & \sqrt{2}\xi & 0 & 0 & -\sqrt{2} & -\sqrt{2}\xi \\ \sqrt{2}\xi & -\sqrt{2} & 0 & 0 & -\sqrt{2}\xi & \sqrt{2} \\ 1 & \xi & -\sqrt{2} & -\sqrt{2}\xi & 1 & \xi \\ \xi & -1 & -\sqrt{2}\xi & \sqrt{2} & \xi & -1 \end{pmatrix}, \tag{146}$$

where $\xi = (1+\sqrt{5})/2$ is the golden ratio. In the sandwich picture, the reference boundary is the canonical one, and the TDLs are in one-to-one correspondence with anyons in the chiral MTC $\mathrm{Spin}(7)_1 \times \overline{\mathrm{Fib}}$ :

$$\eta \leftrightarrow \psi, \mathcal{W} \leftrightarrow \varepsilon, \mathcal{N} \leftrightarrow \sigma. \tag{147}$$

*a. Perturbed by $\sigma'$* Let's consider the $\sigma'$ perturbation of the tricritical Ising model. This perturbation breaks the $\mathbb{Z}_2$ and the $\mathcal{N}$ but preserves $\mathcal{W}$ symmetry. The RG flow is known to arrive at a gapped phase described by a non-trivial TQFT. In the GL picture, this corresponds to an asymmetric double well with two minimum.

Suppose the coupling is positive, in the $\langle W_{\sigma'} \rangle = \sqrt{2} > 0$ phase, we have two Cardy states: $|1\rangle$ and $|\epsilon'\rangle$. From Eq. (141), we know that they are degenerate. Since $\mathcal{W}$ is preserved, we can check the expectation value of the $\mathcal{W}$ in these two degenerate ground states. Noting that $\mathcal{W}$ acts on 1 by $\xi$ and on $\epsilon'$ by $-1/\xi$, we obtain

$$\langle 1|\mathcal{W}|1\rangle = \xi, \ \langle \epsilon'|\mathcal{W}|\epsilon'\rangle = \frac{-1}{\xi}. \tag{148}$$

This is consistent with the property of the non-trivial TQFT described in Ref. [81].

*b. Perturbed by $\epsilon'$* Here we consider the $\epsilon'$ perturbation with negative coupling. The RG flow is known to ends in a first-order transition with three degenerate ground states. In the $\langle W_{\epsilon'} \rangle < 0$ phase, we find that $\langle W_{\epsilon'} \rangle = -1/\xi$ for $|\epsilon'\rangle$, $|\sigma\rangle$, and $|\epsilon\rangle$ and they are degenerate. The perturbation preserves the $\mathbb{Z}_2$ and $\mathcal{N}$ lines. By the action of the topological defect lines given in Ref. [81], we find that $|\epsilon'\rangle$ and $|\epsilon\rangle$ are $\mathbb{Z}_2$ even and $|\sigma\rangle$ is $\mathbb{Z}_2$ odd. Also $\mathcal{N}$ takes the eigenvalues $\pm\sqrt{2}$ for $|\epsilon'\rangle$ and $|\epsilon\rangle$ and annihilates $|\sigma\rangle$. We can also compute the $\mathbb{Z}_2$ order parameter given by $\langle W_\sigma \rangle$:

$$\langle \epsilon'|W_\sigma|\epsilon'\rangle = \frac{-\sqrt{2}}{\xi}, \tag{149}$$

$$\langle \sigma|W_\sigma|\sigma\rangle = 0, \tag{150}$$

$$\langle \epsilon|W_\sigma|\epsilon\rangle = \frac{\sqrt{2}}{\xi}. \tag{151}$$

We see that $|\epsilon'\rangle$ and $|\epsilon\rangle$ are the pair of the spontaneous symmetry-breaking ground states and $|\sigma\rangle$ is the disorder state. Therefore, the gapped phase corresponds to the first-order transition where these three states are degenerate, which is consistent with the GL picture where $\epsilon'$ perturbation corresponds to the $\phi^4$ coupling. Note that if we consider positive coupling, we obtain a similar conclusion but it's actually not correct since it's known that the system flows to the Ising CFT in this case. This is the limitation of this method where it can not distinguish first-order transition v.s. continuous transition.

c. *Perturbed by $\epsilon$* We then consider the $\epsilon$ perturbation which preserves the $\mathbb{Z}_2$ symmetry. In the GL picture, this corresponds to the $\phi^2$ coupling so we expect to see a disorder state on one sign of the coupling and spontaneous symmetry breaking for the other sign of the coupling.

We first consider positive coupling. There are three Cardy states with positive $\langle W_\epsilon \rangle$:

$$\langle 1|W_\epsilon|1\rangle = \xi, \tag{152}$$

$$\langle \sigma|W_\epsilon|\sigma\rangle = 1/\xi, \tag{153}$$

$$\langle \epsilon''|W_\epsilon|\epsilon''\rangle = \xi. \tag{154}$$

Since the expectation values are different, we need to compare their energy by using Eq. (141). We see that the state $|\sigma\rangle$ has the lowest energy as $1/\xi \sim 0.618 < \xi \sim 1,618$. Therefore, we have a unique ground state, which should be the disorder state. We can check the expectation value of the order parameter vanishes by computing $\langle W_\sigma \rangle = 0$.

For negative coupling, there are also three Cardy states with negative $\langle W_\epsilon \rangle$:

$$\langle \epsilon'|W_\epsilon|\epsilon'\rangle = -1/\xi, \tag{155}$$

$$\langle \sigma'|W_\epsilon|\sigma'\rangle = -\xi, \tag{156}$$

$$\langle \epsilon|W_\epsilon|\epsilon\rangle = -1/\xi. \tag{157}$$

Using Eq. (141), we find that $|\epsilon'\rangle$ and $|\epsilon\rangle$ have the lowest energy so that we have two degenerate ground states (note that the one with minimum energy is given by the one having the smallest $|\langle W_\epsilon \rangle|$ due to the negative coupling). The $\mathbb{Z}_2$ symmetry is spontaneously broken since $\langle W_\sigma \rangle$ has opposite eigenvalues in these two states:

$$\langle \epsilon'|W_\sigma|\epsilon'\rangle = \frac{-\sqrt{2}}{\xi}, \tag{158}$$

$$\langle \epsilon|W_\sigma|\epsilon\rangle = \frac{\sqrt{2}}{\xi}. \tag{159}$$

### 3. $\mathbb{Z}_2 \times \mathbb{Z}_2$ spin chains

Here we consider boundary states of several CFTs that can appear in a $\mathbb{Z}_2 \times \mathbb{Z}_2$ spin chain.

a. *$Ising^2$ theory* We begin with two decoupled critical Ising models, which is described by a $Ising^2$ CFT. The Cardy states are labeled by the anyons in the bulk $Ising^2$ topological order. There are four nearby gapped phases of this model, trivial, $\mathbb{Z}_2^a$ SSB, $\mathbb{Z}_2^b$ SSB, and $\mathbb{Z}_2^a \times \mathbb{Z}_2^b$ SSB phases. We discuss the relevant perturbations of these phases and the corresponding boundary states below.

We consider adding the $\psi_1 + \psi_2$ perturbation. Table. II lists $S_{a,\psi_1+\psi_2}/S_{0a}$ for the convenience of calculating the variational energy of the Cardy states. If the coupling is positive, in the $\langle W_{\psi_1} + W_{\psi_2} \rangle > 0$ phase, there is a unique Cardy state $|\sigma_1\sigma_2\rangle$ that minimize the energy. This state has a $\mathbb{Z}_2 \times \mathbb{Z}_2$ symmetry and all the order parameters have vanishing expectation values: $\langle W_{\sigma_1} \rangle = 0$ and $\langle W_{\sigma_2} \rangle = 0$. If the coupling is negative, we are in the $\mathbb{Z}_2^a \times \mathbb{Z}_2^b$ SSB phase with four degenerate Cardy states: $|1\rangle$, $|\psi_1\rangle$, $|\psi_2\rangle$, and $|\psi_1\psi_2\rangle$, among which the order parameters $W_{\sigma_1}$ and $W_{\sigma_2}$ having non-trivial expectation values.

We then consider $\psi_1 - \psi_2$ perturbation. Using table. II, we find that there are two Cardy states $|\sigma_1\rangle$ and $|\sigma_1\psi_2\rangle$ if the coupling is positive. This is a $\mathbb{Z}_2^b$ SSB phase since the order parameters have expectation values $\langle W_{\sigma_1} \rangle = 0$ and $\langle W_{\sigma_1} \rangle = \pm 1$ for $|\sigma_1\rangle$ and $|\sigma_1\psi_2\rangle$, respectively. Similarly, for negative coupling, there are two Cardy states $|\sigma_2\rangle$ and $|\psi_1\sigma_2\rangle$ and the $\mathbb{Z}_2^a$ symmetry is spontaneously broken.

| | $1$ | $\sigma_1$ | $\psi_1$ | $\sigma_2$ | $\psi_2$ | $\sigma_1\sigma_2$ | $\psi_1\psi_2$ | $\sigma_1\psi_2$ | $\psi_1\sigma_2$ |
|---|---|---|---|---|---|---|---|---|---|
| $S_{a,\psi_1+\psi_2}/S_{0a}$ | $2$ | $0$ | $2$ | $0$ | $2$ | $-2$ | $2$ | $0$ | $0$ |
| $S_{a,\psi_1-\psi_2}/S_{0a}$ | $0$ | $-2$ | $0$ | $2$ | $0$ | $0$ | $0$ | $-2$ | $2$ |

TABLE II. The values of $S_{ab}/S_{0b}$ in the Ising$^2$ theory with $b = \psi_1 + \psi_2$ and $b = \psi_1 - \psi_2$

*b.* *Ising$^{2*}$ theory* Another closely related model can be obtained by applying the $\mathbb{Z}_2 \times \mathbb{Z}_2$ SPT entangler to the two decoupled Ising models[62]. We can view it as the stacking of the cluster state and the two critical Ising chains, and make the symmetry act diagonally. The analysis of the boundary states is completely paralleled to the previous two paragraphs. The SPT state is driven by the $\psi_1 + \psi_2$ perturbation with positive coupling. The corresponding Cardy state is $|\sigma_1\sigma_2\rangle$, which is the tensor product of the free boundary condition of the Ising model. The fact that there is only one boundary state with the lowest energy suggests that there will be no ground state degeneracy for an open chain. It was explicitly shown in Ref. [82] that there is no edge degeneracy in this case.

*c.* *Ising$^{2*}$/$\mathbb{Z}_2$ = XY theory* As discussed in Sec. IV A, the transition between the trivial and the SPT phase is described by the $\mathbb{Z}_2$ orbifold of the Ising$^{2*}$ theory, which we denoted as Ising$^{2*}$/$\mathbb{Z}_2$ with the partition function given in Eq. (75). It turns out that this partition function is the same as the XY model[62] (the free Dirac point of the $c = 1$ compact boson CFT). The bulk picture of the Ising$^{2*}$/$\mathbb{Z}_2$ theory is given by a Ising$^2$ topological order defined on an strip with an insertion of a non-invertible defect line. This non-invertible defect line can be viewed as a thin strip of $U(1)_4$ topological order sitting in the middle of the Ising$^2$ theory, where the $U(1)_4$ theory is obtained by the $\psi_1\psi_2$ condensation in the Ising$^2$ theory. The anyons in the $U(1)_4$ theory are labeled by $\{1, \lambda, \bar{\lambda}, \tilde{\psi}\}$ with $\mathbb{Z}_4$ fusion rules:

$$\lambda \times \lambda = \bar{\lambda} \times \bar{\lambda} = \tilde{\psi}, \quad \lambda \times \bar{\lambda} = 1, \quad \tilde{\psi} \times \tilde{\psi} = 1$$
$$\lambda \times \tilde{\psi} = \bar{\lambda}, \quad \bar{\lambda} \times \tilde{\psi} = \lambda. \tag{160}$$

They are mapped to the anyons in the Ising$^2$ theory as follows

$$1 = 1 \oplus \psi_1\psi_2,$$
$$\lambda = \sigma_1\sigma_2,$$
$$\bar{\lambda} = \sigma_1\sigma_2,$$
$$\tilde{\psi} = \psi_1 \oplus \psi_2. \tag{161}$$

There are two confined anyons which are invariant under the condensation:

$$\tilde{\sigma}_1 = \sigma_1 \oplus \sigma_1\psi_2, \quad \tilde{\sigma}_2 = \sigma_2 \oplus \psi_1\sigma_2 \tag{162}$$

The relevant fusion rules are $\tilde{\sigma}_1 \times \tilde{\sigma}_1 = \tilde{\sigma}_2 \times \tilde{\sigma}_2 = 1 + \tilde{\psi}$, and $\tilde{\sigma}_1 \times \tilde{\sigma}_2 = \lambda + \bar{\lambda}$.

We can explicitly check that the partition function of the $U(1)_4$ CFT agrees with Eq. (75). We first extend the region of $U(1)_4$ such that the entire bulk of the strip is given by the $U(1)_4$ topological order. Upon folding the strip, we obtain a sandwich described by $U(1)_4 \boxtimes \overline{U(1)_4}$ with left reference boundary being the canonical boundary, and the right boundary is described by the Ising$^2$ CFT. Using Eq. (4), we obtain the partition function:

$$Z = |\chi_1|^2 + |\chi_\lambda|^2 + |\chi_{\bar{\lambda}}|^2 + |\chi_{\tilde{\psi}}|^2 \tag{163}$$
$$= |\chi_1^2 + \chi_\psi^2|^2 + 2|\chi_\sigma^2|^2 + |2\chi_1\chi_\psi|^2 \tag{164}$$

It is then straightforward to check that Eq. (164) is equal to Eq. (75). The microscopic $\mathbb{Z}_2 \times \mathbb{Z}_2$ symmetry corresponds to the charge conjugation $C$ and the $\mathbb{Z}_2$ symmetry $R$ generated by the $\tilde{\psi}$ line in the $U(1)_4$ CFT.

Now we discuss the boundary states of $U(1)_4$ CFT. The boundary states are in one-to-one correspondence to the particles in the theory $\mathcal{T}$, given by the particles in Eq. (161) and Eq. (162). The Cardy states are given by the deconfined anyons in Eq. (161) in the $U(1)_4$ theory. There are two non-Cardy boundary states given by the confined particles in Eq. (162).

The four Cardy states are listed as follows. The Cardy states $|1\rangle$ and $|\tilde{\psi}\rangle$ corresponds to the $\mathbb{Z}_2^R$ symmetry breaking phases since they are invariant under the charge conjugation symmetry $C$ and exchanged under

fusing the $\tilde{\psi}$ line. The $|\lambda\rangle$ and $|\bar{\lambda}\rangle$ corresponds to the $\mathbb{Z}_2^C$ symmetry breaking phases as they are exchanged under the charge conjugation $C$ and are invariant under the $\mathbb{Z}_2^R$ symmetry.

There are two non-Cardy boundary states $|\tilde{\sigma}_1\rangle$ and $|\tilde{\sigma}_2\rangle$. It is straightforward to see that these two boundary states are invariant under the $\mathbb{Z}_2^C \times \mathbb{Z}_2^R$ symmetry. We may identify them as the SPT and the trivial phases. If we consider an open chain with the two boundary conditions $|\tilde{\sigma}_1\rangle$ and $|\tilde{\sigma}_2\rangle$ at the two ends, the boundary partition function is given by

$$\mathcal{Z}_{\tilde{\sigma}_1 \tilde{\sigma}_2} = \chi_\lambda + \chi_{\bar{\lambda}} = 2\chi_\sigma^2 \tag{165}$$

$$= \frac{\vartheta_2}{\eta} \tag{166}$$

$$= 2q^{\frac{1}{12}}(1 + 2q + 3q^2 + \cdots). \tag{167}$$

where we have used Eq. (138) and the definition of the $\vartheta_2$ and $\eta$ functions

$$\vartheta_2 = \sum_{n \in \mathbb{Z}} q^{\frac{1}{2}(n-\frac{1}{2})^2}, \eta = q^{\frac{1}{24}} \prod_{n=1}^{\infty} (1 - q^n).$$

We find that there is a two-fold degeneracy which is consistent the nontrivial projective representation of the $\mathbb{Z}_2 \times \mathbb{Z}_2$ symmetry.

$U(1)_4$ CFT is a free compact boson theory. We now discuss how to describe the boundary states in the compact boson representation. First, it turns out that these four Cardy states all correspond to the Dirichlet boundary conditions in the free compact boson theory. Since all of our boundary states can be written in terms of the boundary states in the Ising$^2$ CFT, using the dictionary between the boundary states in the Ising$^2$ CFT and the compact boson provided in Ref. [83], we find

$$|1\rangle = |D(0)\rangle, \tag{168}$$

$$|\tilde{\psi}\rangle = |D(\pi)\rangle, \tag{169}$$

$$|\lambda\rangle = |D(\pi/2)\rangle, \tag{170}$$

$$|\bar{\lambda}\rangle = |D(-\pi/2)\rangle, \tag{171}$$

where $|D(\varphi_0)\rangle$ is the Dirichlet boundary condition in the compact boson theory. Note that the charge conjugation symmetry acts on the compact boson $\varphi$ and the dual variable $\theta$ by

$$C : \varphi \to -\varphi, \ \theta \to -\theta, \tag{172}$$

and the $\mathbb{Z}_2^R$ symmetry acts by

$$R : \varphi \to \varphi + \pi, \ \theta \to \theta. \tag{173}$$

Using these symmetry, we can also see that $|D(0)\rangle$ and $|D(\pi)\rangle$ break $\mathbb{Z}_2^R$, and $|D(\pi/2)\rangle$ and $|D(-\pi/2)\rangle$ break $\mathbb{Z}_2^C$.

On the other hand, the two symmetry-preserving boundary statesd correspond to the Neumann boundary conditions in the compact boson:

$$|\tilde{\sigma}_1\rangle = |N(\pi)\rangle, \tag{174}$$

$$|\tilde{\sigma}_2\rangle = |N(0)\rangle. \tag{175}$$

### 4.  Three-state Potts model

Here we consider the boundary states of the three-state Potts model. The three-state Potts model is closely related to the minimal model $\mathcal{M}(6,5)$, which has 10 Virasoso primaries with spins:

$$(1,1)_0, (1,2)_{\frac{1}{8}}, (4,3)_{\frac{2}{3}}, (4,2)_{\frac{13}{8}}, (4,1)_3, (2,1)_{\frac{2}{5}}, (2,2)_{\frac{1}{40}}, (3,3)_{\frac{1}{15}}, (3,2)_{\frac{21}{40}}, (3,1)_{\frac{7}{5}}, \tag{176}$$

Among the ten Virasoro primaries in $\mathcal{M}(6,5)$, only six appear in the energy spectrum of the three-state Potts chain with periodic boundary conditions. They are $(1,1)_0, (2,1)_{\frac{2}{5}}, (3,1)_{\frac{7}{5}}, (4,1)_3, (3,3)_{\frac{1}{15}}$, and $(4,3)_{\frac{2}{3}}$,

| | 0 | 1 | 2 | 3 | 4 |
|---|---|---|---|---|---|
| 1 | 0 | 1/8 | 2/3 | 13/8 | 3 |
| $\tau$ | 2/5 | 1/40 | 1/15 | 21/40 | 7/5 |

TABLE III. The correspondence between $JK_4 \boxtimes$ Fib and the spins. The anyons of $JK_4$ are denoted by $2j$, where $j = 0, \frac{1}{2}, \cdots, 2$

where the subscripts denote the corresponding spins. The torus partition function is a non-diagonal modular invariant:

$$Z = |\chi_{1,1} + \chi_{4,1}|^2 + |\chi_{2,1} + \chi_{3,1}|^2 + 2|\chi_{3,3}| + 2|\chi_{4,3}|. \tag{177}$$

By including the chiral Virasoro primary field $(4, 1)$ in the chiral algebra, the Virasoro algebra is extended to a $W_3$ algebra. The $W_3$ characters are given by

$$\chi_{\mathbb{1}} = \chi_{1,1} + \chi_{4,1}, \tag{178}$$

$$\chi_{\epsilon} = \chi_{2,1} + \chi_{3,1}, \tag{179}$$

$$\chi_{\sigma} = \chi_{\sigma^\dagger} = \chi_{3,3}, \tag{180}$$

$$\chi_{\psi} = \chi_{\psi^\dagger} = \chi_{4,3}. \tag{181}$$

In terms of these $W_3$ characters, the torus partition function is diagonal.

This non-diagonal partition function can also be understood from the bulk TQFT point of view. The topological order corresponding to $\mathcal{M}(6,5)$ can be expressed as the tensor product of the Fibonacci category (i.e. $(G_2)_1$) and $JK_4$, where JK stands for Jones-Kauffman. Here $JK_4$ is a cousin of the more familiar $SU(2)_4$ MTC: they have the same fusion rules, so we label the anyons using $SU(2)$ spins $2j, j = 0, 1/2, 1, 3/2, 2$, but the topological spins are different. For $JK_4$, the spins are

$$\theta_0 = 1, \theta_1 = -\theta_3 = e^{\frac{\pi i}{4}}, \theta_2 = e^{\frac{4\pi i}{3}}, \theta_4 = 1. \tag{182}$$

$SU(2)_4$ has the same spins except $\theta_2 = e^{\frac{2\pi i}{3}}$. Another difference is that the chiral central charge of $JK_4$ is $-2$, while it is 2 for $SU(2)_4$. We denote this category as $JK_4 \boxtimes$ Fib. The correspondence between $JK_4 \boxtimes$ Fib and the $\mathcal{M}(6,5)$ spins is given in Table. III.

The $JK_4 \boxtimes$ Fib has only one non-trivial bosonic condensation, which is condensing the highest spin $j = 2$ particle, denoted by $z$, corresponding to the $W_3$ current. After the $z$ condensation, we have six deconfined anyons:

$$\mathbb{1} = (0, \mathbf{1}) \oplus (4, \mathbf{1}), \quad \epsilon = (0, \tau) \oplus (4, \tau),$$
$$\sigma = (2, \tau)_1, \quad \sigma^\dagger = (2, \tau)_2, \quad \psi = (2, \mathbf{1})_1, \quad \psi^\dagger = (2, \mathbf{1})_2, \tag{183}$$

where we denote the anyons in $JK_4 \boxtimes$ Fib as a pair $(a, b)$. Note that $(1, \mathbf{1})$ and $(1, \tau)$ split in the condensed phase. The $z$ condensed phase is described by the category $\overline{SU(3)_1} \boxtimes$ Fib. The bulk picture of the three-state Potts model is given by a strip of $JK_4 \boxtimes$ Fib with an insertion of a defect, consisted of a thin strip of the $\overline{SU(3)_1} \boxtimes$ Fib theory, in the middle. The $W_3$ characters of the non-diagonal modular invariant are simply labeled by the deconfined anyons in $\overline{SU(3)_1} \boxtimes$ Fib.

Now we discuss the boundary states of the three-state Potts model and the corresponding gapped phases. There are six Cardy states labeled by the deconfined anyons in $\overline{SU(3)_1} \boxtimes$ Fib. First we deform the critical theory by a perturbation $\epsilon$, which leads to ferromagnetic and paramagnetic states. The fixed boundary conditions are $|\mathbb{1}\rangle$, $|\psi\rangle$, $|\psi^\dagger\rangle$ and we may identify them as the $\mathbb{Z}_3$ symmetry-breaking states. The other three Cardy states $|\epsilon\rangle$, $|\sigma\rangle$, $|\sigma^\dagger\rangle$, corresponding to the mixed boundary conditions, are obtained from the symmetry-breaking ones by inserting a $\epsilon$ flux, and thus also break the $\mathbb{Z}_3$ symmetry.

There are two remaining boundary states which are labeled by the confined anyons during the $z$ condensation:

$$\eta = (1, \mathbf{1}) \oplus (3, \mathbf{1}), \quad \xi = (1, \tau) \oplus (3, \tau). \tag{184}$$

The free boundary state $|\eta\rangle$ corresponds to the $\mathbb{Z}_3$ symmetric trivial phase. The remaining boundary state $|\xi\rangle$ is the "new" boundary condition introduced in Ref. [84] and also corresponds to a $\mathbb{Z}_3$ symmetric phase. Boundary partition functions can be readily written down by using Eq. (138).

## 5.   SU(2)$_k$  WZW model

In this section we consider the boundary states of SU(2)$_k$ CFTs. We denote chiral primaries of SU(2)$_k$ theory by $2j$, where $j$ is the $SU(2)$ spin $j = 0, \frac{1}{2}, \cdots, \frac{k}{2}$. The corresponding characters are denoted by $\chi_{2j}(\tau)$.

The modular-invariant partition functions of SU(2)$_k$ CFTs are completely classified, known as the ADE classification [85–87]. Here we discuss the corresponding bulk TQFT picture. The bulk picture of a SU(2)$_k$ CFT is simply a $\mathcal{C} = $ SU(2)$_k$ topological order defined on a strip with an insertion of a defect in the middle. In general, this defect belongs to two classes. For the first class, the defect can be viewed as a very thin strip of another MTC $\mathcal{D}$ separated by a gapped interface $\mathcal{T}$. In all examples below, $\mathcal{D}$ is obtained from $\mathcal{C}$ by condensing an algebra $A$, and $\mathcal{T}$ is a category $\mathcal{C}_A$ of $A$ modules in $\mathcal{C}$. In the second class, the defect is an invertible one, corresponding to a topological symmetry of the SU(2)$_k$ MTC.

The category of $\mathcal{T} = \mathcal{C}_A$ and the condensed theory $\mathcal{D}$ for SU(2)$_k$ topological order are completely classified by the Dynkin diagrams of the ADE type[55, 88]. There are four cases:

1. $A_n$ series of all $k$,corresponding to no defect in the strip.

2. $D_{2n+2}$ for $k = 4n$,

3. $E_6$ for $k = 10$,

4. $E_8$ for $k = 28$.

We briefly summarize the results of Ref. [88] below. There is a correspondence between $\mathcal{T}$ and Dynkin diagrams of types $A_n$, $D_{2n+2}$, $E_6$, $E_8$. Under this correspondence, the algebra $A$ always sit at the longest leg of the diagram, and the simple objects of $\mathcal{T}$ correspond to vertices. Deconfined anyons in $\mathcal{D}$ correspond to filled circles in the Dynkin diagram.

1. $D_{2m+2}$ series for $k = 4m$: Filled circles correspond to the deconfined anyons in $\mathcal{D}$ written in terms of the anyon labels in SU(2)$_k$ theory. Open circles are simple objects in $\mathcal{T}$ but not in $\mathcal{D}$. We see that $A = 0 \oplus 4m$.

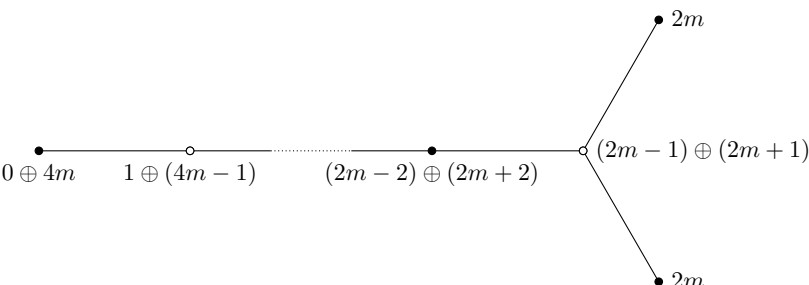

2. $E_6$ for $k = 10$ with $A = 0 \oplus 6$:

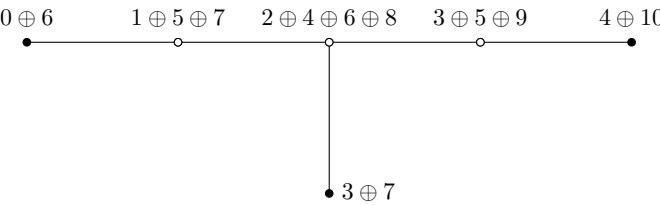

3. $E_8$ for $k = 28$ with $A = \alpha_1$ below:

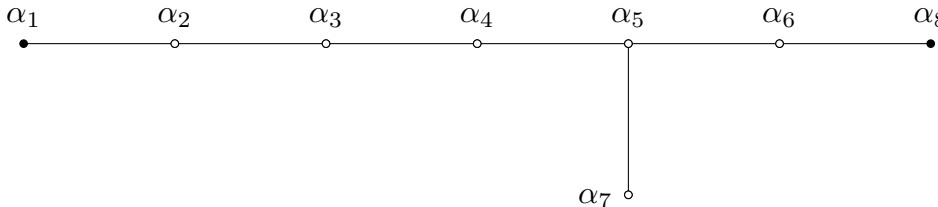

with

$$\alpha_1 = 0 \oplus 10 \oplus 18 \oplus 28$$
$$\alpha_2 = 1 \oplus 9 \oplus 11 \oplus 17 \oplus 19 \oplus 27$$
$$\alpha_3 = 2 \oplus 8 \oplus 10 \oplus 12 \oplus 16 \oplus 18 \oplus 20 \oplus 26$$
$$\alpha_4 = 3 \oplus 7 \oplus 9 \oplus 11 \oplus 13 \oplus 15 \oplus 17 \oplus 19 \oplus 21 \oplus 25$$
$$\alpha_5 = 4 \oplus 6 \oplus 8 \oplus 10 \oplus 12 \oplus 14 \oplus 14 \oplus 16 \oplus 18 \oplus 20 \oplus 22 \oplus 24$$
$$\alpha_6 = 5 \oplus 7 \oplus 11 \oplus 13 \oplus 15 \oplus 17 \oplus 21 \oplus 23$$
$$\alpha_7 = 5 \oplus 9 \oplus 13 \oplus 15 \oplus 19 \oplus 23$$
$$\alpha_8 = 6 \oplus 12 \oplus 16 \oplus 22$$

The modular invariant partition function is then given by the deconfined anyons in $\mathcal{D}$ labeled by the filled circle in the Dynkin diagram. For example, the $E_6$ type modular invariant correspond to the condensate $A = 0 \oplus 6$ in $\mathrm{SU}(2)_{10}$ and the resulting state is $\mathrm{Spin}(5)_1$ topological order. The $\mathrm{Spin}(5)_1$ CFT partition function is

$$Z = |\chi_0 + \chi_6|^2 + |\chi_3 + \chi_7|^2 + |\chi_4 + \chi_{10}|^2. \tag{185}$$

The boundary states are labeled by the simple objects in the module category $\mathcal{T}$, which correspond to the vertices in the Dynkin diagram. Among those boundary states, the Cardy states correspond to the deconfined anyons in $\mathcal{D}$, labeled by the filled circle in the Dynkin diagram. Non-Cardy boundary states are labeled by the open circles, corresponding to the confined anyons. For example, for $\mathrm{Spin}(5)_1$ CFT, there are three Cardy states

$$|\mathbb{1}\rangle = |0 \oplus 6\rangle, \quad |\sigma\rangle = |3 \oplus 7\rangle, \quad |\psi\rangle = |4 \oplus 7\rangle. \tag{186}$$

There are three non-Cardy boundary states:

$$|\eta_1\rangle = |1 \oplus 5 \oplus 7\rangle, \quad |\eta_2\rangle = |2 \oplus 4 \oplus 6 \oplus 8\rangle, \quad |\eta_3\rangle = |3 \oplus 5 \oplus 9\rangle. \tag{187}$$

Boundary partition functions can be obtained by using Eq. (136) or Eq. (138). For example,

$$\mathcal{Z}_{\eta_1,\eta_1} = \chi_{\mathbb{1}} + \chi_{\eta_2} \tag{188}$$
$$= (\chi_0 + \chi_6) + (\chi_2 + \chi_4 + \chi_6 + \chi_8) \tag{189}$$
$$= q^{-5/48}(1 + 3q^{1/6} + 5q^{1/2} + 17q + \cdots), \tag{190}$$

$$\mathcal{Z}_{\mathbb{1},\eta_1} = \chi_1 + \chi_5 + \chi_7 \tag{191}$$
$$= 2q^{-1/24}(1 + 3q^{2/3} + 3q + \cdots), \tag{192}$$

where $q = \exp(-\pi\beta/L)$.

Another physically relevant example is given by condensing $A = 0 \oplus 4$ in $\mathrm{SU}(2)_4$ and obtain an $\mathrm{SU}(3)_1$ topological order, which we have discussed in Sec. V E 4 as a subcategory in the three-state Potts model. The three Cardy states for $\mathrm{SU}(3)_1$ CFT are $|\mathbf{0}\rangle = |0 \oplus 4\rangle$, $|\mathbf{3}\rangle = |2_1\rangle$, $|\bar{\mathbf{3}}\rangle = |2_2\rangle$, where the $j = 1$ particle splits after the condensation. There is only one non-Cardy boundary state from the confined particle: $|\mathrm{free}\rangle = |1 \oplus 3\rangle$, which corresponds to a free boundary condition. It is straightforward to write down the boundary partition function by using Eq. (136) or Eq. (138). For example,

$$\mathcal{Z}_{\mathrm{free},\mathrm{free}} = \chi_{\mathbf{0}} + \chi_{\mathbf{3}} + \chi_{\bar{\mathbf{3}}} \tag{193}$$
$$= \chi_0 + \chi_4 + 2\chi_2 \tag{194}$$
$$= q^{-1/12}(1 + 6q^{1/6} + 8q^{1/2} + 18q^{2/3} + \cdots), \tag{195}$$
$$\mathcal{Z}_{0,\mathrm{free}} = \chi_1 + \chi_3 \tag{196}$$
$$= 2q^{1/24}(1 + 2q^{1/2} + 3q + \cdots). \tag{197}$$

Other boundary partition functions can be obtained similarly.

Lattice realizations of both examples (non-Cardy boundary states for $\mathrm{Spin}(5)_1$ and $\mathrm{SU}(3)_1$) have been studied recently in Ref. [89].

## ACKNOWLEDGMENTS

We thank the authors of Ref. [90] for informing us their work on gapless SPT phases, which has some overlap with our results. We thank Ryohei Kobayashi, Marvin Qi, Hong-Hao Tu and Xiao-Gang Wen for helpful discussions. M.C. would like to acknowledge NSF for support under award number DMR-1846109.

## Appendix A: Review of gapped boundaries and Lagrangian algebra

Here we give a brief review of the gapped boundaries and the Lagrangian algebra. Please see Ref. [42, 51, 79, 88, 91, 92] for more details. An algebra in a MTC $\mathcal{C}$ is a direction sum of of simple objects: $\mathcal{A} = \oplus_\alpha w_\alpha \alpha$, $\alpha \in \mathcal{C}$. More formally, we define a Lagrangian algebra $\mathcal{A}$ in a MTC $\mathcal{C}$ to be an object $\mathcal{A} \in \mathcal{C}$ along with a multiplication morphism $\mu : \mathcal{A} \otimes \mathcal{A} \to \mathcal{A}$ and unit morphism $\iota : 1 \to \mathcal{A}$ such that the following conditions hold:

- Commutativity: $\mu \circ R_{\mathcal{A},\mathcal{A}} = \mu$, where $R_{\mathcal{A},\mathcal{A}}$ is the braiding in $\mathcal{C}$. It can be expressed diagrammatically:

$$
\text{}
\tag{A1}
$$

where solid line represents $\mathcal{A}$ and the junction where three $\mathcal{A}$ lines meet is the morphism $\mu$.

- Associativity: $\mu \circ (\mu \otimes id) = \mu \circ (id \otimes \mu)$,

$$
\text{}
\tag{A2}
$$

- Unit: $\mu \circ (\iota \otimes id) = id$,

$$
\text{}
\tag{A3}
$$

  $\mathcal{A}$ is called connected if $\mathrm{Hom}(1, \mathcal{A})$ is 1-dimensional

- Separability: There exists a splitting morphism $\sigma : \mathcal{A} \to \mathcal{A} \otimes \mathcal{A}$ such that $\mu \circ \sigma = id$, and satisfies

$$
\text{}
\tag{A4}
$$

- Lagrangian: $D_\mathcal{A} = D_\mathcal{C}$, where $D_\mathcal{A} = \sum_{\alpha \in \mathcal{C}} w_\alpha d_\alpha$ is the dimension of the algebra $\mathcal{A} = \oplus_\alpha w_\alpha \alpha$, and $D_\mathcal{C} = \sqrt{\sum_{\alpha \in \mathcal{C}} d_\alpha^2}$ is the total quantum dimension of $\mathcal{C}$.

It can be shown that $\mathcal{A}$ is a commutative associative algebra in a MTC $\mathcal{C}$ if and only if $\mathcal{A}$ decomposes into simple objects as $\mathcal{A} = \oplus_\alpha w_\alpha \alpha$, with $\theta_\alpha = 1$ for all $\alpha$ such that $w_\alpha \neq 0$[93]. The above definition of a Lagrangian algebra is the same as a special, symmetric Frobenius algebra[51, 92], with a Lagrangian condition such that $\mathcal{A}$ has the maximally possible quantum dimension. Physically, it means that we enter a trivially gapped phase after the anyon condensation given by $\mathcal{A}$, or equivalently $\mathcal{A}$ describes the set of anyons that are condensed on the gapped boundary between the topological order $\mathcal{C}$ and the vacuum.

## Appendix B: $\mathbb{Z}_N$ parafermion CFT and the DQCP partition function

The $\mathbb{Z}_N$ parafermion CFT is a diagonal RCFT described by the coset model $\mathrm{SU}(2)_N/\mathrm{U}(1)_{2N}$ with central charge $c = \frac{2(N-1)}{N+2}$ [94–96]. There are $\frac{N(N+1)}{2}$ primaries which are labeled by a pair of integers $(l, m)$ in the range

$$0 \leq l \leq N, \ -l + 2 \leq m \leq l, \ l - m \in 2\mathbb{Z}, \tag{B1}$$

together with the identification

$$(l, m) \sim (N - l, m - N) \sim (N - l, m + N). \tag{B2}$$

The conformal dimensions of the primaries are given by

$$h_{lm} = \bar{h}_{lm} = \frac{l(l+2)}{4(N+2)} - \frac{m^2}{4N}. \tag{B3}$$

The parafermion CFT has a non-anomalous $\mathbb{Z}_N$ symmetry and the primary labeled by $(l, m)$ carries a $\mathbb{Z}_N$ charge $m \bmod N$.

The modular T-matrix is given by

$$T_{lm}{}^{l'm'} = e^{2\pi i h_{lm}} \delta_l{}^{l'} \delta_m{}^{m'} \tag{B4}$$

and the modular S-matrix is

$$S_{lm}{}^{l'm'} = \frac{2}{\sqrt{N(N+2)}} \sin\left[\frac{\pi(l+1)(l'+1)}{N+2}\right] e^{\frac{i\pi mm'}{N}}. \tag{B5}$$

The twisted partition functions is given by[96]

$$Z_{gh} = \sum_{l,m} e^{\frac{2\pi i h(m+g)}{N}} \chi_{l,m+2g} \bar{\chi}_{l,m}. \tag{B6}$$

Now we would like to write the $\mathbb{Z}_2 \times \mathbb{Z}_2$ DQCP partition function in terms of the characters of the $\mathbb{Z}_4$ parafermoin CFT. As shown in Sec. IV C, the DQCT partition function in the $\mathbb{Z}_4$ toric code anyon basis is

$$Z_{\mathrm{DQCP}} = Z_1 + Z_{e^2} + Z_{m^2} + Z_{e^2 m^2}. \tag{B7}$$

In general, the partition functions in the anyon sectors are given by the linear combination of the twisted partition functions. Here we list the relevant twisted partition functions explicitly.

$$
\begin{aligned}
Z_{00} &= Z_1 + Z_e + Z_{e^2} + Z_{e^3} = \sum_{l,m} |\chi_{l,m}|^2, \\
Z_{01} &= Z_1 + iZ_e - Z_{e^2} - iZ_{e^3} = \sum_{l,m} e^{\frac{2\pi i m}{4}} |\chi_{l,m}|^2, \\
Z_{02} &= Z_1 - Z_e + Z_{e^2} - Z_{e^3} = \sum_{l,m} e^{\frac{4\pi i m}{4}} |\chi_{l,m}|^2, \\
Z_{03} &= Z_1 - iZ_e - Z_{e^2} + iZ_{e^3} = \sum_{l,m} e^{\frac{6\pi i m}{4}} |\chi_{l,m}|^2,
\end{aligned}
\tag{B8}
$$

$$Z_{20} = Z_{m^2} + Z_{em^2} + Z_{e^2m^2} + Z_{e^3m^2}$$
$$= \chi_{0,0}\bar{\chi}_{4,0} + \chi_{1,1}\bar{\chi}_{3,1} + \chi_{1,-1}\bar{\chi}_{3,-1} + \chi_{2,2}\bar{\chi}_{2,2} + \chi_{2,0}\bar{\chi}_{2,0}$$
$$+ \chi_{3,1}\bar{\chi}_{1,1} + \chi_{3,-1}\bar{\chi}_{1,-1} + \chi_{4,2}\bar{\chi}_{4,-2} + \chi_{4,0}\bar{\chi}_{0,0} + \chi_{4,-2}\bar{\chi}_{4,2},$$

$$Z_{21} = Z_{m^2} + iZ_{em^2} - Z_{e^2m^2} - iZ_{e^3m^2}$$
$$= -\chi_{0,0}\bar{\chi}_{4,0} - i\chi_{1,1}\bar{\chi}_{3,1} + i\chi_{1,-1}\bar{\chi}_{3,-1} + \chi_{2,2}\bar{\chi}_{2,2} - \chi_{2,0}\bar{\chi}_{2,0}$$
$$- i\chi_{3,1}\bar{\chi}_{1,1} + i\chi_{3,-1}\bar{\chi}_{1,-1} + \chi_{4,2}\bar{\chi}_{4,-2} - \chi_{4,0}\bar{\chi}_{0,0} + \chi_{4,-2}\bar{\chi}_{4,2},$$

$$Z_{22} = Z_{m^2} - Z_{em^2} + Z_{e^2m^2} - Z_{e^3m^2}$$
$$= \chi_{0,0}\bar{\chi}_{4,0} - \chi_{1,1}\bar{\chi}_{3,1} - \chi_{1,-1}\bar{\chi}_{3,-1} + \chi_{2,2}\bar{\chi}_{2,2} + \chi_{2,0}\bar{\chi}_{2,0}$$
$$- \chi_{3,1}\bar{\chi}_{1,1} - \chi_{3,-1}\bar{\chi}_{1,-1} + \chi_{4,2}\bar{\chi}_{4,-2} + \chi_{4,0}\bar{\chi}_{0,0} + \chi_{4,-2}\bar{\chi}_{4,2},$$

$$Z_{23} = Z_{m^2} - iZ_{em^2} - Z_{e^2m^2} + iZ_{e^3m^2}$$
$$= -\chi_{0,0}\bar{\chi}_{4,0} + i\chi_{1,1}\bar{\chi}_{3,1} - i\chi_{1,-1}\bar{\chi}_{3,-1} + \chi_{2,2}\bar{\chi}_{2,2} - \chi_{2,0}\bar{\chi}_{2,0}$$
$$+ i\chi_{3,1}\bar{\chi}_{1,1} - i\chi_{3,-1}\bar{\chi}_{1,-1} + \chi_{4,2}\bar{\chi}_{4,-2} - \chi_{4,0}\bar{\chi}_{0,0} + \chi_{4,-2}\bar{\chi}_{4,2}.$$

$$(B9)$$

Note that the $\mathbb{Z}_4$ symmetry is implemented by the $m$ line on the left boundary, which can be pulled into the $m$ line in the bulk.

Using Eq. (B7), Eq. (B8), and Eq. (B9), we obtain the partition function for the $\mathbb{Z}_2 \times \mathbb{Z}_2$ DQCP:

$$Z_{\mathrm{DQCP}} = \frac{1}{2}\left(Z_{00} + Z_{02} + Z_{20} + Z_{22}\right)$$
$$= |\chi_{0,0}|^2 + 2|\chi_{2,2}|^2 + 2|\chi_{2,0}|^2 + |\chi_{4,2}|^2 + |\chi_{4,0}|^2 + |\chi_{4,-2}|^2 \qquad (B10)$$
$$+ \chi_{0,0}\bar{\chi}_{4,0} + \chi_{4,0}\bar{\chi}_{0,0} + \chi_{4,2}\bar{\chi}_{4,-2} + \chi_{4,-2}\bar{\chi}_{4,2}. \qquad (B11)$$

This is the orbifold of the $\mathbb{Z}_2$ center symmetry in the diagonal $\mathrm{SU}(2)_4/\mathrm{U}(1)_8$ CFT. It corresponds to choosing the $D$ type modular invariant in the parent $\mathrm{SU}(2)_4$ theory. It is known that the diagonal $\mathbb{Z}_4$ parafermion $\mathrm{SU}(2)_4/\mathrm{U}(1)_8$ CFT can be represented as the $\mathrm{U}(1)_6/\mathbb{Z}_2$ CFT. Therefore, the DQCP transition can also be represented as the $\mathrm{U}(1)_6$ CFT[94, 96].

---

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
