# Peer review of "Topological holography, quantum criticality, and boundary states"

_SciPost Physics_

## Round 2 · Referee Report · Anonymous (Referee 1) · 2024-3-5

Strengths

  1. Clear presentation of the physical picture and results;
  2. Excellent review of the literature, covering both gapped and gapless phases (including critical points);
  3. Provides a unified framework to describe both gapped and gapless phases, enabling non-perturbative statements about phases, phase transitions, and dualities;
  4. Detailed and careful analysis;

Weaknesses

  1. the paper is a bit too long without a clear emphasis at the beginning of each section;
  2. a lot of gapped and gapless phases (or critical points) are discussed. However, there is no summary provided at the end, which makes it challenging for readers to grasp the merit of the paper.

Report

The author attempts to provide a unified description of exotic gapped/gapless phases and unconventional phase transitions in 1+1d, which is termed topological holography. This approach describes the generalized global symmetry of a local quantum system in terms of a topological order in one higher dimension.

In my opinion, this is an excellent paper and undoubtedly interesting to the community of high energy and condensed matter, deserving publication. However, I do have the following comments that should be clarified before publication:

  1. The phase transition from SSB to SPT (protected by symmetry G) phases belongs to symmetry-enriched quantum critical points. The authors provide a topological holography description of the symmetry-enriched Ising CFTs (G=Z2 X Z2, discrete group) in the main text. Although the authors also mention the SU(2)_{k} WZW (continuous group case) in "Sec.V Boundary states in RCFTs". I wonder if for the case where G is a continuous group (such as U(1) or SU(2) et al.), can we also use the picture of topological holography to characterize the symmetry-enriched G CFT?

2.The bulk-boundary correspondence in topological holography reminds me of the Li-Haldane conjecture, which states that 1+1d CFT has a one-to-one correspondence with the 2+1d chiral (gapped) topological phase. For the gapped topological phases, can topological holography provide a more intuitive understanding of the Li-Haldane conjecture? (In this work (arXiv:2112.05886), the authors claims wormhole picture can provide a intuitive understanding of the Li-Haldane conjecture, and further generalized for systems beyond gapped topological phases) Moreover, gapless SPT or symmetry-enriched quantum critical points, as mentioned by the author in the main text, has attracted much attention in the past few years. These gapless phases also exhibit universal bulk-boundary correspondence, as discussed in paper arxiv:2402.04042. Similarly, in addition to the boundary CFT analysis provided by the above paper, can topological holography provide a more intuitive understanding of the universal entanglement spectrum in gapless SPT states? I hope the author can engage in some simple discussions on the above issues.

3.The author has developed a solid topological holographic picture for (1+1)d gapped or gapless quantum phases. Can this picture be generalized to higher dimensions? The author should briefly discuss this issue.

4.The author attempts to develop a unified theoretical description, at least for 1+1d, of conventional and unconventional (beyond Landau) phase transition theory. This effort reminds me of the topological order and unconventional phase transitions mentioned in arXiv:2204.03045, which can be understood as higher-form symmetry breaking and thus may fall within the framework of the generalized Landau paradigm. What are the connections and differences between the topological holography description and generalized Landau paradigm presented in arXiv: 2204.03045? I hope the author can provide a brief discussion on these issues.

---

## Round 2 · Referee Report · Anonymous (Referee 2) · 2024-4-11

Strengths

  1. The manuscript developed the symmetry topological field theory to critical system and gapless phases. To the best of my knowledge, this is the first work on applying the SymTFT to gapless SPTs.
  2. The manuscript discussed computation of conformal boundary conditions from SymTFT. This investigation appears to be the first of its kind to apply SymTFT to the analysis of conformal boundary conditions.
  3. The manuscript proposed a unified framework for studying topological aspects of 1+1D gapless systems and presented the application of this framework to multiple examples. This work advances our understanding of symmetry-enriched gapless systems, marking a substantial contribution to the field.

Weaknesses

  1. There is not a general theory of topological holograph for gapless systems. For instance, the discussion on gapless SPT focuses on a single example and lacks a general story.

  2. The presentation of this work can be improved. Specifically, there are some geometry setup with no figure illustration, which may cause difficulty in understanding the derivation. For instance, the "inserting a pair of domain wall" process on page 11 is not easy to picture without explicit illustration. The term"the other side" is not defined. It is also unclear where these domain walls are located.

Report

I believe this work meets the publication criteria for Sci-Post. It is among the first studies to apply symmetry topological field theory (SymTFT) to gapless symmetry-protected topological states (gSPTs) and deconfined quantum criticality (DCQC), as well as to explore conformal boundary conditions using SymTFT. The authors showed via many concrete computations how the partition function and topological response of 1+1D gapless systems can be extracted from their holographic dual. This research advances our understanding of the topological properties of gapless systems and establishes SymTFT as a robust tool for analyzing the topological aspects of symmetry in gapless systems.

Requested changes

I suggest adding figure illustration for the "inserting domain wall" process on page 11.

Recommendation

Publish (easily meets expectations and criteria for this Journal; among top 50%)

---

## Round 2 · Referee Report · Anonymous (Referee 3) · 2024-4-13

Strengths

Please see the report.

Weaknesses

Please see the report.

Report

A lot of this paper is a review of formalism developed by Wen and collaborators and by Freed, Moore and Telemann.
There have been many papers about this formalism and in my opinion the value it adds is not yet clear.
The present paper would be more useful if it were written for a reader who was skeptical of the value of this point of view, that is, for someone who may not care about the development of this formalism for its own sake.

The authors use the name "topological holography" for this construction. I think the term "holographic" is quite misleading in this context. Unlike in the context which is usually called holography (or holographic duality or AdS/CFT) in theoretical physics, there is no map of Hilbert spaces here. The bulk is being used as an auxiliary device to describe the boundary, placed there in addition to the system. I don't think the word "holographic" needs to be even further overloaded. This is only my opinion, but it is strongly felt.

The main new outcome in this paper seems to be that the authors can use this formalism to predict the partition functions (and hence operator content) of the 1+1d critical theories separating various 1+1d gapped phases, in terms of Virasoro characters. Some such results have appeared sporadically in the literature, but the present paper provides a nice unified perspective. This seems like a nice set of results that should be published.

There is also a set of statements about 1+1d CFT boundary states as domain walls between the CFT and gapped phases.
I have to admit that I was not able to understand what is the sharp statement that the authors extract here about such boundary states.
The fact that there is a relation between relevant perturbations and boundary states is not new. The claim seems to be that the authors provide a "holographic interpretation" of conformal boundary states.
This "interpretation" is explained mainly in Figure 9, which I was not able to absorb.
Perhaps the figure could be clarified somehow.
But even if I were able to undersatnd it, did not yet understand the value of such an interpretation. So I think this discussion should be sharpened and clarified.

-- My impression is that the results about relations between boundary states and gapped phases have some overlap with the paper
https://arxiv.org/abs/2210.01135
which should at least be cited.

-- In the very important list of results:
"We demonstrate through many examples that the correspondence between the topological gapped boundary conditions in the sandwich picture and the (1+1)d gapped phases." is not a complete sentence. What do they demonstrate about this correspondence?

-- page 3: In the sentence "In this paper, we work with the anyon basis in contrast to the basis labeled by the flat connections. "
basis for what? At this point in the text, it is not clear what vector space is being discussed.

-- page 4:
"There is a canonical choice for the basis states |α⟩. When M is a torus,..."
I didn't understand. Are the authors saying that there is a canonical basis for any $M$ or just when $M$ is a torus?

-- Ref 53 has only an author list.

-- page 14: "comes naturally with the following datum" . "datum" is singular.

-- there is a broken link in ref 77.

-- The fact that many examples are provided is nice.

Requested changes

Please see the report.

Recommendation

Ask for major revision

---

## Round 3 · Referee Report · Xue-Jia Yu (Referee 1) · 2025-4-15

Report

The reply and subsequent additions to the manuscript fully answer my
concerns. I recommend publication of the manuscript to SciPost Physics.

Recommendation

Publish (surpasses expectations and criteria for this Journal; among top 10%)

---

## Round 3 · Referee Report · Anonymous (Referee 3) · 2025-5-4

Report

I apologize for the delay. I made the mistake of trying to go through the whole (very long) paper again. The paper should be published. Below are a few typos I found.

-- first paragraph: "long-rang"

-- after equation (4) : of of
"with an insertion of of Lagrangian algebra "

-- in the paragraph before equation (7): an local
"each condensation channel i corresponds to an local operator"

Recommendation

Publish (meets expectations and criteria for this Journal)

---

## Round 3 · Author Response

Report 1:

“In my opinion, this is an excellent paper and undoubtedly interesting to the community of high energy and condensed matter, deserving publication.”

Reply: We are pleased that the referee for the positive feedback regarding our work and the recommendation for publication. We have added a conclusion section as suggested by the referee.

“A lot of gapped and gapless phases (or critical points) are discussed. However, there is no summary provided at the end, which makes it challenging for readers to grasp the merit of the paper.”

Reply: We thank the referee for the suggestion. We have added a conclusion section to summarize our results.

“1. The phase transition from SSB to SPT (protected by symmetry G) phases belongs to symmetry-enriched quantum critical points. The authors provide a topological holography description of the symmetry-enriched Ising CFTs (G=Z2 X Z2, discrete group) in the main text. Although the authors also mention the SU(2)k WZW (continuous group case) in ”Sec.V Boundary states in RCFTs”. I wonder if for the case where G is a continuous group (such as U(1) or SU(2) et al.), can we also use the picture of topological holography to characterize the symmetry-enriched G CFT?”

Reply: We thank the referee for bringing up the questions about continuous symmetry groups. There are several proposals on the generalization of topological holography to include continuous symmetry groups from the field theory perspective. Although the complete framework is still under development, we believe it is possible to study symmetry-enriched quantum critical points by using topological holography and this will be an interesting future direction.

“2. The bulk-boundary correspondence in topological holography reminds me of the Li-Haldane conjecture, which states that 1+1d CFT has a one-to-one correspondence with the 2+1d chiral (gapped) topological phase. For the gapped topological phases, can topological holography provide a more intuitive understanding of the Li- Haldane conjecture? (In this work (arXiv:2112.05886), the authors claims wormhole picture can provide a intuitive understanding of the Li-Haldane conjecture, and further generalized for systems beyond gapped topological phases) Moreover, gapless SPT or symmetry-enriched quantum critical points, as mentioned by the author in the main text, has attracted much attention in the past few years. These gapless phases also exhibit universal bulk-boundary correspondence, as discussed in paper arxiv:2402.04042. Similarly, in addition to the boundary CFT analysis provided by the above paper, can topological holography provide a more intuitive understanding of the universal entanglement spectrum in gapless SPT states? I hope the author can engage in some simple discussions on the above issues.”

Reply: Currently, we are not sure if topological holography can provide further insight about the Li-Haldane conjec- ture but it is an interesting question to think about. Regarding the gapless SPT states, we believe that it is possible to use topological holography to obtain boundary states of the gapless SPT states. The correspondence between the entanglement spectrum and the edge spectrum should still hold. This is an interesting future direction.

“3. The author has developed a solid topological holographic picture for (1+1)d gapped or gapless quantum
phases. Can this picture be generalized to higher dimensions? The author should briefly discuss this issue.”

Reply: Yes, the topological holographic picture can be generalized to higher dimensions, and it is a very important direction. In higher dimensions, the symmetry is described by a fusion n-category, which characterizes the boundary topological defects/operators of some bulk TQFT. The formal mathematical framework has been developed by Freed, Moore, and Telemann in Ref. 31, but many details still need to be spelled out explicitly in order to apply it to the study of phases and phase transitions.

“The author attempts to develop a unified theoretical description, at least for 1+1d, of conventional and
unconventional (beyond Landau) phase transition theory. This effort reminds me of the topological order and unconventional phase transitions mentioned in arXiv:2204.03045, which can be understood as higher-form symmetry breaking and thus may fall within the framework of the generalized Landau paradigm. What are the connections and differences between the topological holography description and generalized Landau paradigm presented in arXiv: 2204.03045? I hope the author can provide a brief discussion on these issues.”

Reply: Topological holography is a unified picture to study the generalized Landau paradigm. More specifically,
the topological defect lines living on the reference boundary implement the higher fusion category symmetry, which for example include the higher-form, higher group, and non-invertible symmetries. One can use topological holography to study phases and phase transitions for systems with the higher fusion category symmetry. We include a brief discussion in the conclusion section.

Report 2:

“I believe this work meets the publication criteria for Sci-Post. It is among the first studies to apply symmetry topological field theory (SymTFT) to gapless symmetry-protected topological states (gSPTs) and deconfined quantum criticality (DCQC), as well as to explore conformal boundary conditions using SymTFT. The authors showed via many concrete computations how the partition function and topological response of 1+1D gapless systems can be extracted from their holographic dual. This research advances our understanding of the topological properties of gapless systems and establishes SymTFT as a robust tool for analyzing the topological aspects of symmetry in gapless systems.”

Reply: We are pleased that the referee recommends our manuscript for publication and sincerely thank for the positive feedback regarding our work.

“I suggest adding figure illustration for the ”inserting domain wall” process on page 11.”

Reply: We thank the referee for pointing out the potential confusion. The discussion on page 11 has been simplified and rewritten for clarity. We remove the argument based on ”inserting domain wall” and the new argument only use the anyon condensation.

Report 3:

“The main new outcome in this paper seems to be that the authors can use this formalism to predict the partition functions (and hence operator content) of the 1+1d critical theories separating various 1+1d gapped phases, in terms of Virasoro characters. Some such results have appeared sporadically in the literature, but the present paper provides a nice unified perspective. This seems like a nice set of results that should be published.”

Reply: We thank the referee for carefully reading our paper and the positive feedback, and for the recommendation for publication.

“There is also a set of statements about 1+1d CFT boundary states as domain walls between the CFT and gapped phases. I have to admit that I was not able to understand what is the sharp statement that the authors extract here about such boundary states. The fact that there is a relation between relevant perturbations and boundary states is not new. The claim seems to be that the authors provide a ”holographic interpretation” of conformal boundary states. This ”interpretation” is explained mainly in Figure 9, which I was not able to absorb. Perhaps the figure could be clarified somehow. But even if I were able to understand it, did not yet understand the value of such an interpretation. So I think this discussion should be sharpened and clarified.”

Reply: We thank the referee for the question. While conformal boundary states can be studied purely within BCFT, it is always beneficial to explore multiple approaches, as a challenging question in one framework may be more tractable in another.

The key purpose of this section is to develop an alternative framework—topological holography—for studying conformal boundary states. Specifically, we show that these states can be interpreted as coupling edge chiral CFTs to form a thin torus with different anyon flux insertions. The resulting states on the thin torus naturally correspond to the ground states of a (1+1)d gapped phase.

From this perspective, we derive the boundary partition functions in Eq. (133) and Eq. (136). In particular, we believe Eq. (136) provides a new formula for the boundary partition functions of non-diagonal RCFTs.

To enhance clarity, we have revised Section V.B and updated the captions of Fig. 9 and Fig. 10. Our work demonstrates that topological holography successfully reproduces known results on conformal boundary states for RCFTs, with numerous examples provided. Moreover, we believe this framework could be extended to study boundary states of more exotic gapless phases, such as intrinsically gapless SPT phases, which is an intriguing direction for future research. We have added this potential application into the conclusion section.

“My impression is that the results about relations between boundary states and gapped phases have some overlap with the paper https://arxiv.org/abs/2210.01135 which should at least be cited.”

Reply: We thank the referee for pointing out this paper. This paper is related to the discussion about the boundary states for transition between the SPT and trivial phases in our work. We have cited this paper in the revision.

“In the very important list of results: ”We demonstrate through many examples that the correspondence between the topological gapped boundary conditions in the sandwich picture and the (1+1)d gapped phases.” is not a complete sentence. What do they demonstrate about this correspondence?”

Reply: We thank the referee for pointing out the potential confusion. They demonstrate the correspondence between gapped boundary conditions and (1+1)d gapped phases, given a fixed reference boundary. The correspon- dence is obtained by analyzing the order parameters and disorder operators in the sandwich picture. We have revised it in the new version.

“- page 3: In the sentence ”In this paper, we work with the anyon basis in contrast to the basis labeled by the flat connections. ” basis for what? At this point in the text, it is not clear what vector space is being discussed.”

We thank the referee for pointing out the potential confusion. We have removed this sentence and merged it into section II. A.

“- page 4: ”There is a canonical choice for the basis states. When M is a torus,...” I didn’t understand. Are the authors saying that there is a canonical basis for any M or just when M is a torus?”

There is a canonical choice for the basis states for any manifold M. We have added a reference (Ref. 50), which discusses the choice of basis on a general manifold explicitly.

“- Ref 53 has only an author list.
- page 14: ”comes naturally with the following datum” . ”datum” is singular. 
- there is a broken link in ref 77.”

We thank the referee for pointing out the reference error and typos. We have corrected these in the revision.

We thank the referee for raising these insightful questions. We hope that we have adequately addressed the questions concerning our work. We hope that the referee will find our response sufficient and that the current version of our draft will be judged suitable for publication in SciPost Physics.

---

## Round 3 · List of Changes

1. A conclusion section is added.

2. The abstract is revised to also emphasize on gapped phases.

3. On page 3, the sentence ”In this paper, we work with the anyon basis in contrast to the basis labeled by the flat connections. ” is removed and merged it into section II. A.

4. The discussion on page 11 is simplified without using any “domain wall insertion”.

5. Typos are corrected.

---

## Editorial Decision

in_voting